# Metal-organic framework template-guided electrochemical lithography on substrates for SERS sensing applications

Youyou Lu [1,2], Xuan Zhang [3], Liyan Zhao[2], Hong Liu[2], Mi Yan[2,4], Xiaochen Zhang[1], Kenji Mochizuki [3] ✉ & Shikuan Yang [1,2,4,5] ✉

The templating method holds great promise for fabricating surface nanopatterns. To enhance the manufacturing capabilities of complex surface nanopatterns, it is important to explore new modes of the templates beyond their conventional masking and molding modes. Here, we employed the metal-organic framework (MOF) microparticles assembled monolayer films as templates for metal electrodeposition and revealed a previously unidentified guiding growth mode enabling the precise growth of metallic films exclusively underneath the MOF microparticles. The guiding growth mode was induced by the fast ion transportation within the nanochannels of the MOF templates. The MOF template could be repeatedly used, allowing for the creation of identical metallic surface nanopatterns for multiple times on different substrates. The MOF template-guided electrochemical growth mode provided a robust route towards cost-effective fabrication of complex metallic surface nanopatterns with promising applications in metamaterials, plasmonics, and surface-enhanced Raman spectroscopy (SERS) sensing fields.

Metallic nanopatterns play essential roles in the fields of plasmonics, sensing, metamaterials, and biology[1–3]. Various methods have been developed to fabricate metallic surface nanopatterns, including electron-beam and photolithography[4,5], microcontact or printing[6,7], and imprinting[8]. However, these methods often suffer from high cost, require specialized equipment, involve multiple complex and time-consuming steps, or rely on aggressive chemistry. In contrast, the self-assembly of nanospheres allows for the rapid formation of large-area ordered monolayer nanosphere films within seconds[9,10]. By utilizing self-assembled monolayer nanosphere array templates, a cost-effective technique called colloidal lithography has been developed to synthesize metallic surface nanopatterns over large areas[11–13]. These monolayer nanosphere templates can function as masks during the thermal evaporation of metallic films, enabling the creation of bowtie antenna surface nanopatterns[14]. They can also act as molds during

the electrodeposition growth of metals, resulting in the formation of nanobowl arrays[11,12,15,16] (Fig. 1a). Notably, the masking and the molding mode of the monolayer nanosphere templates limit the structural diversity of the resulting surface nanopatterns[13]. Therefore, exploring new functionalities of self-assembled templates is important to enhance their capabilities of designing metallic surface nanopatterns.

Metal-organic frameworks are a kind of crystalline materials composed of coordination bonds formed between metal ions and multidentate organic linkers, forming interconnected nanochannels that can be precisely tailored in size[17–19]. Various simple and high-throughput chemical processes have been developed to prepare well-defined morphologies of MOF microparticles[20–22], enabling them to self-assemble into three-dimensionally or two-dimensionally ordered structures[23–25]. As an analog, nanochannels within the cell membranes play a vital role in the selective transportation of desired ions and,

[1]Department of Medical Oncology, The First Affiliated Hospital, Zhejiang University School of Medicine, Hangzhou 310003, China. [2]School of Materials Science and Engineering, Zhejiang University, Hangzhou 310058, China. [3]Department of Chemistry, Zhejiang University, Hangzhou 310058, P. R. China. [4]Baotou Research Institute of Rare Earths, Baotou 014030, China. [5]State Key Laboratory of Fluid Power and Mechatronic Systems, Zhejiang University, Hangzhou 310027, China. ✉e-mail: kenji_mochizuki@zju.edu.cn; shkyang@zju.edu.cn

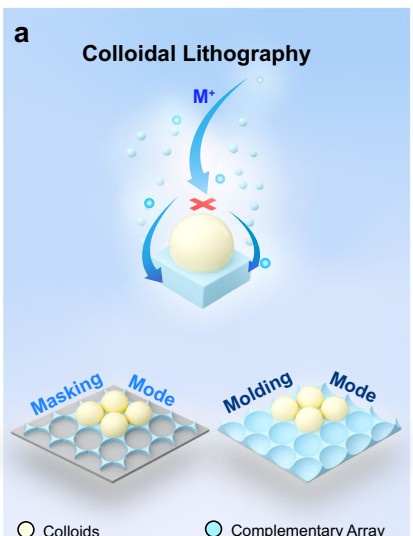
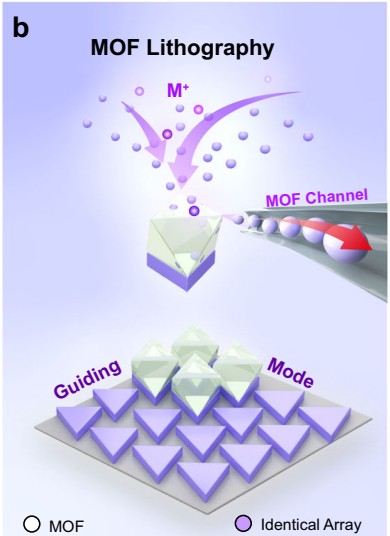

**Fig. 1 | Schematic of the masking and molding growth modes in conventional colloidal lithography and the guiding growth mode in MOF lithography. a** In conventional colloidal lithography, two common modes are observed: masking growth mode and molding growth mode. The colloidal templates screen the metal clusters formed during thermal or electron beam lithography, resulting in the formation of a honeycomb array of nanotriangles (the masking growth mode). The colloidal templates occupy the space during the electrochemical growth of metals from the substrate, leading to the formation of a complementary array to the colloidal templates (the molding growth mode). **b** In MOF template-guided lithography, electrodeposited metals grow precisely underneath the MOF microparticles, rendering the formation of metallic surface nanopatterns with a structure the same as that of the interface of the MOF template and the underneath substrate (the guiding growth mode).

most importantly, in turn, directly influence the biological processes and disease developments[26,27]. In contrast, current researches on MOFs primarily focus on their separation of ions or gases based on their size compatibility[28–30]. Worrall, et al. tried to reduce the metal ions within the nanochannels of the MOF microparticles to prepare metal nanoclusters[31]. We envision that immediate electrochemical reduction of the metal ions selectively passing through the nanochannels within the MOF microparticles can create new surface nanopatterns different from those obtained by conventional colloidal lithographic methods.

Here, we discover a previously unidentified guiding growth function of the MOF template capable of directing the electrochemical growth of metals exclusively underneath the MOF template, giving rise to the formation of metal surface nanopatterns exactly the same as the MOF/substrate interface area (Fig. 1b). The MOF template can be recycled for multiple times. We expect that the MOF template-guided growth mode and the recyclability of the MOF template can greatly strengthen the capability of the colloidal lithography method to repeatedly design complex metallic surface nanopatterns with promising applications in plasmonics and SERS sensing fields, especially after considering the ability to create fine nanopatterns of MOFs using sophisticated electron beam lithographic methods[32,33].

## Results

### Fabrication of monolayer MOF microparticle templates

UiO-66 is a kind of chemically stable MOF and can be easily prepared in different morphologies and sizes. For example, the edge size of the UiO-66 octahedra could be adjusted in the size range from <100 nm to > 100 μm according to previous publications[34,35]. Therefore, we chose to assemble the UiO-66 octahedra into a monolayer membrane to show the MOF template-guided electrochemical growth of metallic surface nanopatterns. We employed a solvothermal method to synthesize well-defined UiO-66 octahedra using acetic acid (AA) as a modulator[23]. The prepared UiO-66 octahedra were well-dispersed in water (Fig. 2a). The X-ray diffraction (XRD) pattern confirmed the octahedra were UiO-66 (Fig. 2b). As examples, the UiO-66 octahedra with a mean edge size of 650 nm, 900 nm, and 1150 nm with a narrow

size distribution (relative standard deviation: <8%) were prepared (Fig. 2c and Supplementary Fig. 1). The pore sizes within the UiO-66 octahedra were determined by a density functional theory method based on nitrogen adsorption-desorption isotherm profiles (Supplementary Fig. 2). The UiO-66 contained small cages with a diameter of 0.6 nm, large cages with a diameter of 0.9 nm[36,37], and pores with a wide size distribution centered at 1.5 nm caused by missing linkers and clusters due to the substitution of AA by 1,4-benzene-dicarboxylate (BDC)[34] (Fig. 2d). The interconnected cages formed a string-like structure with the small cages and the triangular windows acting as the choke points at regular intervals.

We employed an air/water interfacial self-assembly process to prepare monolayer UiO-66 octahedron templates (Fig. 2e and f). In brief, liquid droplets formed by UiO-66 octahedron aqueous dispersions and ethanol were dropped onto a hydrophilic glass slide placed in the center of a petri dish filled with water. The water surface just covered the glass slide. The UiO-66 octahedra were trapped at the water surface and spread out to cover the whole water surface as the droplets were continuously applied onto the glass slide (Supplementary Movie 1). The spreading process of the UiO-66 octahedron was driven by the surface tension gradient known as the Gibbs-Marangoni effect[38]. We lifted up the monolayer UiO-66 octahedra by adding water into the petri dish. The monolayer UiO-66 octahedra floating at the water surface exhibited vibrant iridescence, indicating their ordered arrangement (Fig. 2g). The freestanding monolayer UiO-66 octahedron film could be transferred onto any desired substrates by picking it up from the bottom. All UiO-66 octahedra within the monolayer array used their {111} triangular facets to contact with the substrate proven by the prominent diffraction peak originating from the {111} facets and the disappearance of the diffraction peak of the {002} facets in the XRD pattern (Fig. 2b and h). Self-assembly of spherical polystyrene spheres is easier compared to that of the UiO-66 octahedra because the spheres can easily adjust their orientation to achieve a densely packed monolayer film. In contrast, the octahedra interact with their neighbors through their side faces in an antiparallel manner[39], making the orientation adjustment difficult. We discovered that introducing some

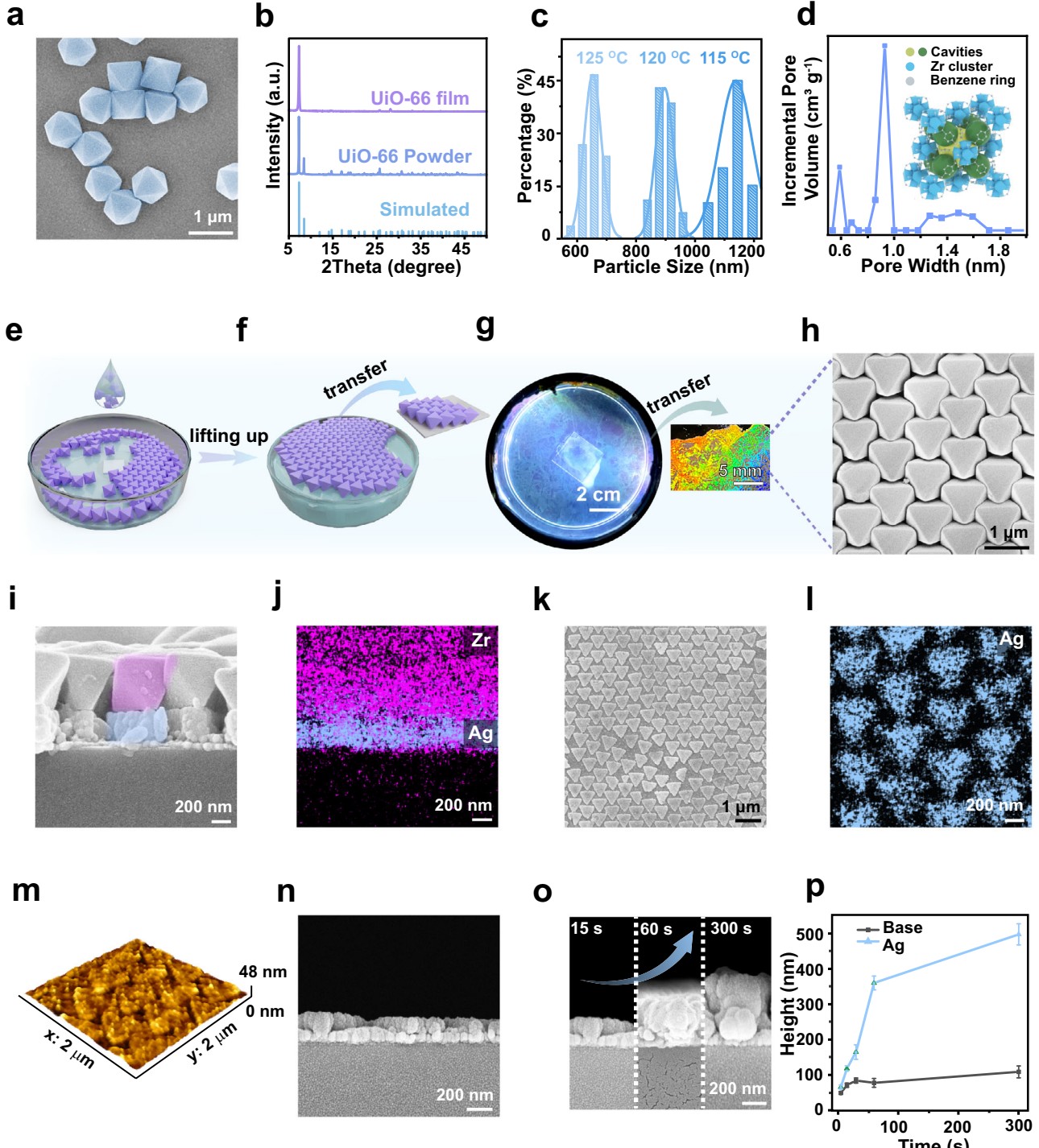

**Fig. 2 | Monolayer UiO-66 octahedron template-guided electrochemical lithography. a** SEM image of the synthesized UiO-66 octahedra. **b** XRD of the UiO-66 octahedron powders and the monolayer UiO-66 octahedron template. **c** Edge size distribution of the UiO-66 octahedra prepared at different temperatures. **d** Pore size distribution of the nanochannels within the UiO-66 octahedra. Inset: the crystallographic model of UiO-66. **e, f** Schematics of the fabrication process of the monolayer UiO-66 octahedron template. **g** The monolayer UiO-66 octahedron template floating at the water surface. The UiO-66 octahedron monolayer template can be transferred onto arbitrary substrates by picking it up from the bottom. **h** SEM image of the monolayer UiO-66 octahedron template. **i, j** Side view of and the element mapping of the 5-min electrodeposited Ag nanotriangle array

underneath the monolayer UiO-66 octahedron template, respectively. **k–m** SEM image of, element mapping of, and atomic force microscope image of the prepared Ag nanotriangle array electrodeposited for 15 s, respectively. **n** Side view of the Ag nanotriangle array. **o** The thickness of the Ag nanotriangles increased as the electroposition proceeded. **p** Thickness increase of the Ag nanotriangles and the nanofilms deposited on bare gold surface (Base) as a function of the electrodeposition time. Error bars represent the standard deviation of ten measurements at different places. The samples were prepared in the electrolyte composed of 300 mM AgNO$_3$ and 14 mM SDS under 1.2 V at different times. Source data are provided as a Source Data file.

ethanol into the octahedron aqueous dispersions at a volume ratio of 4:6 could create an appropriate surface tension gradient and generate a highly ordered monolayer octahedron template. Deviating from this volume ratio resulted in the formation of an octahedron template with multiple layers or large cracks during the assembly process (Supplementary Fig. 3).

## MOF template-guided electrochemical lithography

Ag electrodeposition was used as an example to demonstrate the MOF template-guided electrochemical lithography concept using a two-electrode system. A monolayer UiO-66 octahedron film was transferred onto a piece of gold-covered silicon wafer and used as the working electrode. Surprisingly, the Ag was electrodeposited underneath the UiO-66 octahedra and the electrodeposited Ag lifted up the octahedra, instead of the commonly expected growth within the interstitials of the colloidal crystal template[11–16] (Fig. 2i, j and Supplementary Fig. 4). After removing the UiO-66 octahedra by ultrasonic treatment in water, a hexagonally arranged Ag nanotriangle array was observed with their tips pointing towards the middle of the edges of adjacent nanotriangles (Fig. 2k, l). Tiny Ag nanoparticles appeared on the Ag nanotriangles electrodeposited for 15 s, while the surface of the Ag nanotriangles became smoother after electrodeposition for 300 s (Fig. 2m and Supplementary Fig. 5). The structure of the Ag nanotriangle array completely inherited the interface pattern between the monolayer UiO-66 octahedron template and the underneath substrate. The thickness of the Ag nanotriangles increased quickly at the first 100 s electrodeposition and then slowly increased (Fig. 2o, p). The growth rate of the Ag nanotriangles was about 6.52 nm s⁻¹ on average, while the growth rate of Ag on the bare gold substrate was only ~ 0.67 nm s⁻¹ (Fig. 2p and Supplementary Fig. 6).

## Growth mode diagram as a function of $C_{AgNO3}$ and $C_{SDS}$

The Ag nanotriangle arrays could be prepared in a wide potential range using either a two-electrode or a three-electrode system, or under a constant current electrodeposition mode (Supplementary Figs. 7–9). In contrast, the concentration of Ag⁺ ions ($C_{AgNO3}$) and sodium dodecyl sulfate (SDS) ($C_{SDS}$) within the electrolyte could switch the function of the UiO-66 octahedron template (Fig. 3a, b and Supplementary Fig. 10). When $C_{AgNO3} > 150$ mM and $C_{SDS} > 7$ mM, the monolayer UiO-66 octahedron template played the guiding growth function, resulting in the formation of Ag nanotriangle arrays (Region I in Fig. 3a, b). In a sharp contrast, the monolayer UiO-66 octahedron template switched to the molding mode as $C_{AgNO3} < 150$ mM and $C_{SDS} > 1.4$ mM, resulting in the formation of nanoframe arrays (i.e., complementary to the nanotriangle pattern created under the guiding growth mode) (Region II in Fig. 3a, b). When $C_{SDS} < 1.4$ mM, large blocks with irregular shapes were formed on the top of the monolayer UiO-66 octahedron template (Region III in Fig. 3a, b). In short, the guiding growth mode of the UiO-66 octahedron template needs high $C_{SDS}$ and high $C_{AgNO3}$. At the edge area of Region I, the nanotriangles were not flat with many defects (Fig. 3a and Supplementary Fig. 10). At the edge area of Region II, Ag were electrodeposited both underneath and surrounding the UiO-66 octahedra. Without SDS in the electrolyte, the electrodeposited Ag formed large blocks on the top of the monolayer UiO-66 octahedron template. Therefore, SDS played an important roles in refining the electrodeposited Ag grains.

## MOF template-guided growth mechanism

It is critical to figure out the mechanism behind the unique guiding electrochemical growth mode of the monolayer UiO-66 octahedron template. The opening size of the nanochannels (~ 0.6 nm) within UiO-66 is larger than the size of the Ag⁺ ions (~ 0.3 nm) but smaller than the size of SDS (~ 0.6 nm in width and ~1.8 nm in length)[39,40]. Therefore, Ag⁺ ions can theoretically pass through the nanochannels within the UiO-

66 octahedra, while SDS cannot enter the nanochannels. The hydrophobic monolayer UiO-66 octahedron template exhibited a water contact angle of 122.8° (Supplementary Fig. 11). Introducing SDS into the electrolyte could greatly lower its surface tension, promoting the electrolyte to wet the monolayer UiO-66 octahedron template and facilitating the Ag⁺ ions to enter the nanochannels of the UiO-66 octahedra. We measured the conductivity of the electrolyte before and after adding SDS. SDS resulted in a conductivity reduction of approximately 12% induced by the binding of Ag⁺ ions with the dodecyl sulfate micelles. In contrast, the conductivity of the pressed UiO-66 octahedron pellet filled with the electrolyte dramatically increased by 52% after introducing SDS into the electrolyte (Supplementary Figs. 12 and 13). The conductivity improvement of the UiO-66 octahedron pellet was because of two aspects: 1) the increased wettability of the electrolyte to the UiO-66 pellet and 2) the reduction of the electrostatic repulsion forces during the entering process of the Ag⁺ ions into the Ag⁺ ions filled nanochannels[41] induced by the negatively charged dodecyl sulfate ions weakly bound to the Ag⁺ ions.

To prove that the Ag⁺ ions prefer to enter the nanochannels of the UiO-66 octahedra, we immersed the UiO-66 octahedra into the electrolyte composed of silver nitrate. The Ag⁺ ions continuously entered the UiO-66 octahedra and an absorption equilibrium was reached after about 7 h (Supplementary Fig. 14). After adding additional silver nitrate into the electrolyte, more Ag⁺ ions moved into the UiO-66 octahedra until saturation adsorption achieved or all of the nanochannels were filled with Ag⁺ ions (Fig. 3c).

The difference in chemical potentials ($\Delta\mu$) determines the mass transfer direction. Therefore, we calculated the $\Delta\mu$ of Ag⁺ ions between within UiO-66 nanochannels and in bulk electrolyte solutions theoretically (Fig. 3d). The chemical potential ($\mu$) of Ag⁺ ions was decomposed into the ideal term and the excess term ($\mu^{ex}$):

$$\mu = RT\ln\Lambda^3\rho + \mu^{ex} \tag{1}$$

Where $k_B$ is the Boltzmann constant; $\Lambda$ is the de Broglie thermal wavelength of Ag; $\rho$ is the number density of Ag⁺ ions. Then, $\Delta\mu$ could be illustrated as:

$$\Delta\mu = k_B T\ln\frac{\rho_{UiO-66}}{\rho_{sol}} + \mu^{ex}_{UiO-66} - \mu^{ex}_{Sol} \tag{2}$$

We computed the $\mu^{ex}_{UiO-66}$ to be -454.6 ± 2.3 kJ mol⁻¹ and $\mu^{ex}_{Sol}$ to be -429.7 ± 0.1 kJ mol⁻¹ via molecular dynamics (MD) simulations combined with the Bennett acceptance ratio method (see Method section for details). The difference of $\mu^{ex}_{UiO-66}$ and $\mu^{ex}_{Sol}$ was -24.9 kJ mol⁻¹. The negative value of the difference indicated that Ag⁺ ions preferred to enter the nanochannels of UiO-66 until $\frac{\rho_{UiO-66}}{\rho_{sol}}$ approaching approximately $1.2 \times 10^4$ where $\Delta\mu$ equaled 0. However, we must also take into account the capacity limit of the UiO-66 octahedra. Our MD simulations showed that the population of Ag⁺ ions was highly concentrated at the eight tetrahedral cages and two octahedral cages in the unit cell of UiO-66 with a volume of 9.3 nm³, corresponding to a number density of ~ 1.1 nm⁻³. Thus, when the UiO-66 was immersed into the electrolyte, all of the cages within the UiO-66 octahedra were occupied by the Ag⁺ ions with a saturated number density $\rho^{sat}_{UiO-66}$ of ~ 1.1 nm⁻³. The saturated number density of Ag⁺ ions was estimated to be ~ 1.3 nm⁻³ from the adsorption equilibrium (Fig. 3c), which was in good agreement with the MD simulation predicted value. Therefore, both the experimental and the simulation results suggested that all of the cages within the UiO-66 octahedra were occupied by Ag⁺ ions in the electrolyte we used.

We proposed that the moving path of the Ag⁺ ions determined the function of the UiO-66 monolayer template. When the Ag⁺ ions prefer to pass through the UiO-66 octahedra via the inside nanochannels, the Ag⁺ ions will be immediately reduced into Ag atoms

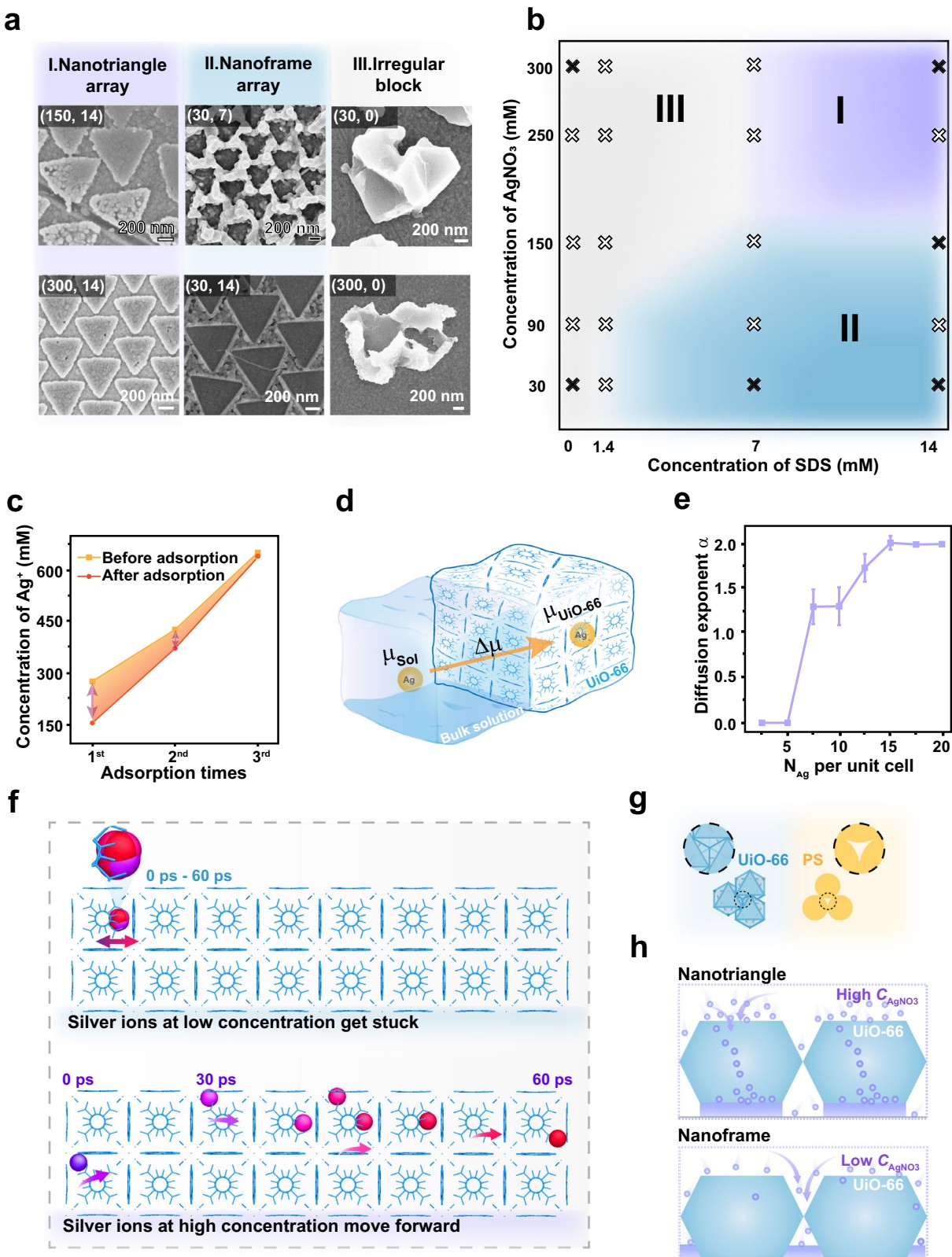

once they contact the underneath electrode surface and gradually form the nanotriangles. In constrast, Ag tends to grow surrounding the UiO-66 octahedra to form the nanoframe array if the Ag+ ions prefer to pass through the monolayer UiO-66 octahedron template via the bulk electrolyte route existing at the defects (*e.g.*, cracks) of the octahedron template. Whether the Ag+ ions choose the nanochannels within the UiO-66 or the solution to pass through the UiO-

66 octahedron monolayer should be influenced by $C_{AgNO_3}$ and $C_{SDS}$ according to the experimental results.

We further performed the MD simulations to investigate the transportation behavior of the Ag+ ions within the nanochannels in the UiO-66 octahedra under an external electric field. We applied a power-law form[40] to describe the one-dimensional diffusion of Ag+ ions in the nanochannels. The growth of the mean squared displacement

**Fig. 3 | Mechanism of the MOF template-guided electrochemical lithography.**
**a** Three kinds of typical nanostructures obtained in the electrolyte composed of different concentrations of AgNO$_3$ and SDS. The numbers in the corner of each image indicated the concentrations of AgNO$_3$ and SDS ($C_{AgNO_3}$, $C_{SDS}$). **b** Three different concentration regions corresponding to the creation of the nanotriangle array, the nanoframe array, and the irregular blocks. **c** The concentration of Ag$^+$ ions remaining in the electrolyte after adsorption by the UiO-66 powder.
**d** Schematic of the chemical potential difference $\Delta\mu$ pushed the Ag$^+$ ions to enter the UiO-66 octahedra. **e** The diffusion exponent $\alpha$ as a function of Ag$^+$ ion numbers

per unit cell of UiO-66. Error bars represent the standard deviation computed from three independent simulations with different initial configurations. **f** Trajectories of a single Ag$^+$ ion in 60 ps (the period of time for Ag$^+$ ions to complete a cycle) at high and low concentrations. The snapshot interval was 10 ps. Water molecules and the other ions were not shown. **g** Schematics of the monolayer UiO-66 octahedron template without through pores and the interstitials formed within the monolayer PS spheres. **h** Moving path of the Ag$^+$ ions at high and low concentrations corresponding to the formation of the nanotriangle array and the nanoframe array, respectively. Source data are provided as a Source Data file.

(MSD) of Ag$^+$ ions over time was described by Equation 3 (Supplementary Fig. 15).

$$\langle|\mathbf{x}(t) - \mathbf{x}(0)|^2\rangle = K_\alpha \cdot t^\alpha \qquad (3)$$

Where $\mathbf{x}(t)$ denotes the $\mathbf{x}$-coordinate of an individual Ag$^+$ ion at time $t$; $K_\alpha$ represents the generalized diffusion constant and $\alpha$ represents the diffusive exponent. The $\mathbf{x}$-axis is oriented perpendicularly to the Ag nanotriangles. The variation of $\alpha$ in a wide range as a function of the Ag$^+$ ion concentration indicates different transportation mechanisms at different Ag$^+$ ion concentrations within the UiO-66 octahedra (Fig. 3e). At low Ag$^+$ ion concentrations (*e.g.*, the number of Ag$^+$ ions in each unit cell is less than five), $\alpha$ approaches zero, reflecting that the Ag$^+$ ions show an oscillatory diffusion behavior. In this case, the low electrostatic repulsion forces among Ag$^+$ ions limit their hopping, leading to negligible diffusion among nanocages (Fig. 3f and Supplementary Movie 2)[42]. In contrast, at high Ag$^+$ ion concentrations, the MSD exhibits a finite slope with $\alpha > 1$. When $\alpha$ value is greater than 1, the superdiffusion takes place, while an $\alpha$ value of 1 corresponds to the Fickian diffusion[43,44]. At high Ag$^+$ ion concentrations, Coulomb interactions between Ag$^+$ ions become prominent and the energy barriers among neighboring nanocages decrease due to the accumulation of positive Ag$^+$ ions confined within the narrow nanochannels. As a result, Ag$^+$ ions rapidly hop from one nanocage to another (Fig. 3f and Supplementary Movie 3). The reduction in the activation energy barrier resulting from multi-ion concerted migration commonly observed in nanopore transportation systems also contributed to the fast Ag$^+$ ion transportation within the nanochannels of the UiO-66 octahedra[45–47].

In addition to the Ag$^+$ ions' diffusion tendency into and fast transportation within the UiO-66 octahedra, the structure of the monolayer UiO-66 octahedron template also played an important role in the MOF template-guided growth mode. Different from the conventional colloidal crystal template formed by polystyrene (PS) spheres with an interstitial within three adjacent PS spheres, no through pores existed within the monolayer UiO-66 octahedron template (Fig. 3g and Supplementary Fig. 16). The Ag$^+$ ions have to enter the UiO-66 octahedra to reach the cathode electrode surface. At high $C_{AgNO_3}$, Ag$^+$ ions enter the UiO-66 octahedra driven by the chemical potentials and move rapidly within the nanochannels. When the Ag$^+$ ions reach the bottom of the UiO-66 octahedra, they are reduced to Ag atoms and eventually form Ag nanotriangles (Fig. 3h). Whereas at low $C_{AgNO_3}$, the Ag$^+$ ions are trapped inside the nanochannels. In this case, the Ag$^+$ ions prefer to pass through the thin sandglass-shaped area to reach the electrolyte trapped by the monolayer UiO-66 octahedron template and the substrate, giving rise to the formation of Ag nanoframe arrays (Fig. 3h).

## Plasmonic properties and SERS sensing applications of the Ag nanotriangle arrays

We first simulated the plasmonic properties of the Ag nanotriangle arrays using the finite-difference time-domain (FDTD) methods (Supplementary Fig. 17). Four absorption peaks appeared at about 450 nm, 530 nm, 550 nm, and 750 nm. The 450 nm and the 550 nm localized surface plasmon (LSPR) peak possibly originated from the

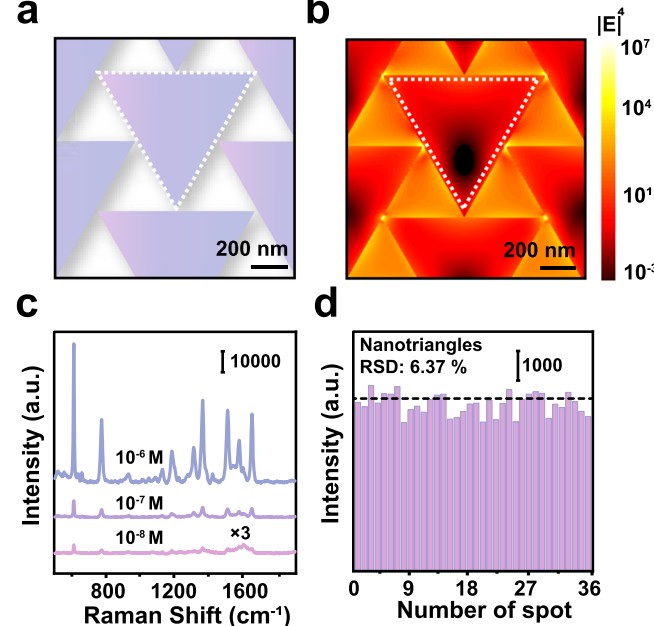

**Fig. 4 | SERS performance of the Ag nanotriangle arrays. a** Schematic of the Ag nanotriangle array. **b** FDTD simulated electromagnetic field distribution over the Ag nanotriangle array excited by a 532 nm laser. **c** SERS spectra of R6G molecules at different concentrations on the Ag nanotriangle array. **d** The intensity of the 612 cm$^{-1}$ SERS peak at 36 randomly chosen sites on the Ag nanotriangle array. Source data are provided as a Source Data file.

quadrupolar and the dipolar mode of individual Ag nanotriangles, while the 530 nm and the 750 nm peak arose from the plasmonic coupling between neighboring nanotriangles in the array[48,49]. The experimentally measured absorption spectra showed two broad peaks at around 550 nm and 740 nm. The discrepancies between the FDTD simulated and the experimentally measured plasmonic properties were induced by structural deviation from the triangular shape and the defects in the ordered array.

Strong electromagnetic fields are located at the edges and the tips of the Ag nanotriangles according to the FDTD simulation results (Fig. 4a, b). These regions with strong electromagnetic fields could behave as SERS hot spots to significantly enhance the Raman signals of chemicals[50,51]. We used Rhodamine 6 G (R6G) as a model molecule to evaluate the SERS sensing performance of the Ag nanotriangle arrays. SERS signals of R6G molecules even at a concentration of 10 nM could be clearly observed (Fig. 4c). The relative standard deviation (RSD) of the 612 cm$^{-1}$ SERS peak intensity was estimated to be 6.37% (Fig. 4d), reflecting a good signal reproducibility.

## Extension to other MOFs and metals

The MOF template-guided growth mode was also observed using the monolayer MIL-96 microparticle template (Fig. 5). Truncated hexagonal bipyramid (THB) shaped MIL-96 microparticles with an average size of 4 μm were synthesized (Fig. 5a–c). The MIL-96 microparticles

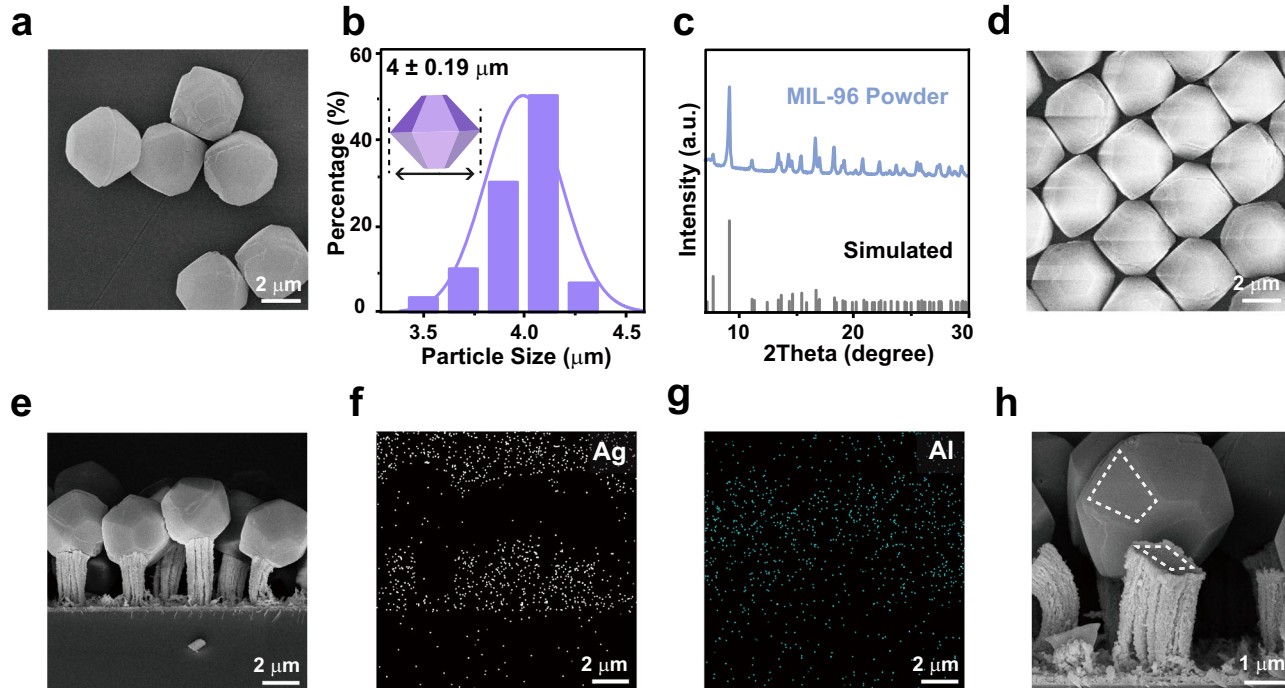

**Fig. 5 | MIL-96 THB-guided electrochemical lithography. a** SEM image of the synthesized MIL-96 THBs. **b** Size distribution of MIL-96 THBs. **c** XRD pattern of the MIL-96 powders. **d** SEM image of the self-assembled monolayer MIL-96 microparticle template. **e–g** Side view of and the element mapping of the Ag micropillar array formed underneath the monolayer MIL-96 THB template after electrodeposition in the electrolyte composed of 30 mM AgNO₃ and 7 mM SDS for 5 min, respectively. **h** A MIL-96 THB detached from the underneath Ag micropillar. The dotted polygons indicates the same shape of the Ag micropillar and the MIL-96 facets. Source data are provided as a Source Data file.

consist nanopores with a diameter of 0.9 nm and estimated opening sizes of 0.3 nm[52]. The monolayer MIL-96 THB template was obtained using the same assembling method used to assemble UiO-66 octahedra (Fig. 5d and Supplementary Fig. 18). The electrodeposited Ag micropillars lifted up the MIL-96 THBs by even several micrometers (Fig. 5e). Element mapping of Ag and Al further verified the relative positions of the Ag micropillars and the MIL-96 THBs (Fig. 5f, g). The Ag nanopillars inherited the trapezoidal shape of the THB's side facets (Fig. 5h). During the template removal process by the ultrasonic treatment in water, the Ag micropillars were easily fractured (Supplementary Fig. 19). To test the versatility of the MOF template-guided electrochemical growth mode, we tried to prepare Cu nanotriangle arrays as another example using the monolayer UiO-66 octahedron template. As expected, Cu nanotriangles formed by nanoparticles were prepared using the monolayer UiO-66 octahedron template (Supplementary Fig. 20).

**Recyclability of the MOF template**
Different from the conventional colloidal lithography technique where the colloidal templates are one-time use, the UiO-66 octahedron monolayer template can be used for multiple times. As an example, we used polybenzimidazole (PBI) to fix the "ZJU" letters formed by closely packed UiO-66 octahedra on a piece of glass slide because of its chemical stability, high mechanical strength, and good ion conductivity[52,53] (Fig. 6a, b). The thickness of the UiO-66@PBI composite film is important because too thick will make it difficult to tightly attach to the substrate, while too thin will be easily broken during the transferring process (Supplementary Fig. 21). The UiO-66@PBI film was detached from the glass slide by soaking it in 90 °C hot water for 24 h. Then, the UiO-66@PBI composite film was transferred onto a piece of gold-coated silicon wafer. After electrodeposition of Ag, "ZJU" letters formed by Ag nanotriangles were created underneath the UiO-66 octahedra (Supplementary Fig. 22). The

adhesion force between the UiO-66@PBI film and the Ag nanotriangles was significantly weaker than that on the glass slide, because of the reduced interface area. Therefore, the UiO-66@PBI template could be easily peeled off from the underneath Ag nanotriangle patterns simply by immersing into ethanol and water in sequence. The detached UiO-66@PBI film would float at the water surface, which could be transferred onto another substrate and used to electrodeposit Ag nanotriangle patterns again (Fig. 6c).

The UiO-66@PBI template can be used to electrochemically print complex surface patterns. As an example, the UiO-66 octahedra were patterned to form the complex logo of Zhejiang University using a mask (Fig. 6d). Then, the pattern was fixed using the PBI film. The PBI film enclosed the top part of the UiO-66 octahedra (Fig. 6e). The UiO-66@PBI logo was translated into Ag nanotriangles formed one after Ag electrodeposition (Fig. 6d, f). Similarly, the UiO-66@PBI film was detached from the Ag nanotriangles by soaking it in ethanol and water in sequence and ready for reuse (Fig. 6d and Supplementary Fig. 23). MOFs could not only be fabricated into various faceted nanoparticles but could be manufactured into fine nanopatterns using sophisticated lithography methods[25,32]. We anticipate electrochemically print metallic surface nanopatterns using these fine MOF nanopatterns working under the guiding growth function. The recyclability of the MOF nanopatterns can greatly reduce the fabrication time and cost of the metallic surface nanopatterns.

In summary, we discovered the unrevealed guiding growth function of the MOF template capable of electrochemically growing metallic surface nanopatterns exclusively underneath the MOF template. The guiding growth mode was enabled by the fast ion transportation within the nanochannels of the MOF template proven by both the experimental measurements and the MD simulation results. We could intentionally switch from the guiding to the molding growth mode by adjusting the concentration of Ag⁺ ions and SDS in the electrolyte. The prepared Ag nanotriangle patterns show interesting

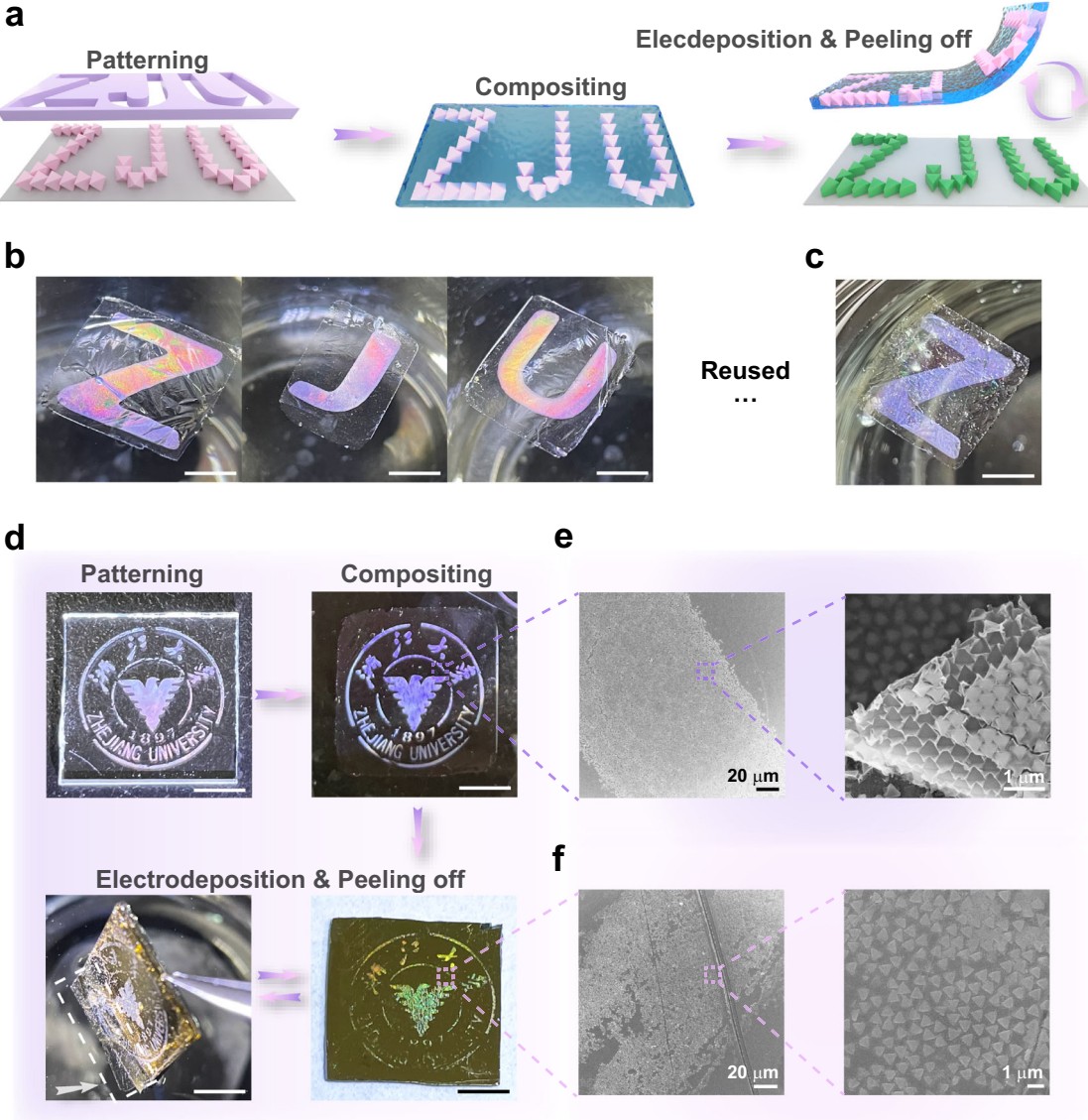

**Fig. 6 | Recyclability of the MOF template in electrodepositing complex Ag surface nanopatterns. a** Schematic of the fabrication and recyclability process of the UiO-66@PBI template. **b** Photos of "Z", "J", and "U" letters patterned by monolayer UiO-66 octahedra within the PBI film. **c** Photo of the UiO-66@PBI template after using twice. **d** Recycling process of the UiO-66@PBI template. Patterning the UiO-66 octahedra into the logo of Zhejiang University on a piece of glass slide. Cementing the pattern using the PBI membrane. Performing Ag electrodeposition using UiO-66@PBI template. Peeling off the UiO-66@PBI template from the Ag nanotriangle array for recycling. Scale bar in all the photos: 5 mm. **e** SEM image of the patterned UiO-66 octahedra within the PBI film. **f** SEM image of the electrodeposited Ag nanotriangle pattern.

plasmonic properties and promising applications in SERS sensing fields. The MOF template-guided electrochemical growth could be extended into other MOFs and the preparation of other metallic surface nanopatterns. More importantly, the MOF template could be easily peeled off from the underneath metallic surface nanopatterns and be used again to fabricate the same metallic surface nanopatterns. The capability to manufacture MOF nanopatterns using sophisticated lithographic methods and their recyclability endow the MOF template with great potentials to create complex metallic surface nanopatterns in a low cost, high throughput, and time-saving manner.

## Methods
### Materials
Zirconium (IV) chloride (ZrCl$_4$, 99.9%), 1,3,5-benzenetricarboxylic acid (BTC, 98.0%) were purchased from Aladdin. Aluminum (III) nitrate nonahydrate (Al(NO$_3$)$_3$·9H$_2$O, 99.0%), Cupric (II) nitrate trihydrate (Cu(NO$_3$)$_2$·3H$_2$O, 99.0%), terephthalic acid (99.0%), acetic acid (99.8%),

dimethylformamide (DMF, 99.5%), methanol (99.5%), ethanol (AR), nitric acid (HNO$_3$, 65%), and silver nitrate (AR) were purchased from Sinopharm Chemical reagent. Sodium dodecyl sulfate (99.0%) was purchased from Innochem. Polybenzimidazoles (98.0%) and N-methylpyrrolidone (NMP, 99.5%) were purchased from Macklin Biochemical.

### Synthesis of the UiO-66 octahedra and MIL-96 THBs
Monodisperse UiO-66 octahedra were synthesized according to previous reports with some modifications[25]. Typically, 350 mg of ZrCl$_4$ and 249 mg of BDC were dissolved into 100 ml of DMF. The solutions were divided into five glass vials with an equal volume. After adding 2.73 ml of AA into each vial, they were placed in an oven at 120 °C for 12 h. The products were centrifuged and washed twice with DMF and methanol, respectively. The collected UiO-66 octahedra were re-dispersed in deionized water at a concentration of 200 mg ml$^{-1}$.

Monodisperse MIL-96 THBs were synthesized as follows. Typically, 3.75 g of $Al(NO_3)_3 \cdot 9H_2O$ was dissolved into 34.3 ml of deionized water before adding 5.7 ml of AA. 2.1 g of BTC was dissolved into 10 ml of DMF. The salt solutions and the BTC solution were mixed at a volume ratio of 4:1 in a Teflon-lined autoclave. The autoclave was heated at 130 °C for 24 h. The MIL-96 microparticles were separated from the reacting solution by centrifugation and re-dispersed in deionized water at a concentration of 200 mg ml$^{-1}$.

### Assembling the UiO-66 octahedra and the MIL-96 THBs into monolayer films

A piece of glass slide (24 mm × 24 mm) and a petri dish (12 cm in diameter and 8 cm in depth) were treated with oxygen plasma (Harrick Plasma, PDC-32G) for 10 min to make the surface super hydrophilic. The UiO-66 octahedron aqueous suspensions at a concentration of 200 mg ml$^{-1}$, deionized water, and ethanol were mixed at a volume ratio of 40:20:40 by ultrasonic treatment for 10 min. Then, the plasma-treated glass slide was placed at the bottom of the petri dish. Deionized water was slowly added into the petri dish until almost covered the glass slide. Subsequently, 100 μl of the mixed solutions composed of UiO-66 octahedra were dropped onto the glass slide. The UiO-66 octahedra were trapped at the water surface and a monolayer was gradually formed at the water surface. Deionized water was slowly filled into the petri dish via a plastic tube to lift up the monolayer UiO-66 octahedron template floating at the water surface. The monolayer UiO-66 octahedron template was transferred onto a 50 nm-thick gold-covered silicon substrate (10 mm × 15 mm) by picking up the template from the bottom. A stainless-steel mask was attached onto the gold-covered silicon substrate before picking up the template to create UiO-66 octahedron patterns. The same self-assembly process was used to assemble MIL-96 THBs into a monolayer film.

### Fabrication of the UiO-66@PBI templates

First, PBI powders were dissolved into NMP at a concentration of 0.2 wt.%. The solution was stirred at 70 °C for 0.5 h. 150 μl of the solutions were dropped onto the glass slide with a patterned UiO-66 octahedron monolayer. After drying in vacuum at 80 °C for 3 h, the glass slide was stored in deionized water at 90 °C for 24 h. The UiO-66@PBI composite film was detached from the glass slide and floated on the water surface. The gold-covered silicon substrate was used to pick up the floating UiO-66@PBI composite film. After the electrodeposition of Ag, the UiO-66@PBI template was easy to be detached from the electrodeposited Ag nanotriangles underneath the template. Simply immersing the sample into deionized water and ethanol alternately for several times, the UiO-66@PBI template was peeled off and floated at the water surface for second-time usage.

### Ag and Cu electrodeposition using the MOF templates

The electrodeposition of Ag was carried out in a two-electrode system unless otherwise stated. In this system, the cathode electrode was the monolayer MOF microparticle template-covered gold substrate and the counter electrode was a carbon rod. Different amounts of $AgNO_3$ and SDS were dissolved into deionized water as the electrolyte. The electrodeposition was performed at 45 °C under a deposition voltage of 1.2 V at different times. A three-electrode setup was also used to prepare the Ag nanotriangle arrays. $Hg/Hg_2SO_4$ was used as a reference electrode.

The electrodeposition of Cu was carried out in a two-electrode setup. The electrolyte was formed by 1200 mM of $Cu(NO_3)_2 \cdot 3H_2O$ and 14 mM SDS. $HNO_3$ was added to adjust the pH to 1. The deposition was carried out at 45 °C under a voltage of 1.8 V for 180 s.

### Material characterizations

XRD patterns were performed to characterize the crystal structure of the MOF microparticles and the monolayer MOF microparticle template (GIXRD, Rigaku D/MAX 2550). The morphology of the MOF microparticles and the electrodeposited metallic surface nanopatterns was investigated using a field-emission scanning electron microscope (SEM) (Hitachi, SU-8010) operating at 15.0 kV equipped with an X-ray energy-dispersive spectrometer. The $N_2$ adsorption/desorption isothermal curves of the UiO-66 octahedra were obtained from the Micromeritics ASAP 2020 surface area analyzer with the pore size analyzing capability operating in liquid argon. The pore size distributions of the nanochannels within the UiO-66 octahedra were simulated by the density functional theory method. The surface tension of the electrolyte and the contact angle of the monolayer UiO-66 octahedron film were determined by the pendent drop method using an automatic contact angle equipment and the contour analysis system, respectively (DataPhysics Instruments, OCA 20). Inductively coupled plasma-mass spectrometry (ICP-MS) was performed to determine the remaining amount of the Ag element in the solution (PerknElmer, NexION 300). The extinction spectra were measured using an UV–Vis–NIR absorption spectrometer (Lambda 950, PerkinElmer).

### Adsorption experiments

We measured the concentration variation of the Ag$^+$ ions before and after adding UiO-66 powders by ICP-MS. 0.1 g UiO-66 powders were added in 1 ml of the electrolyte composed of 300 mM $AgNO_3$ and 14 mM SDS. After 7 h, the concentration of Ag$^+$ ions in the electrolyte remained almost unchanged, indicating the adsorption equilibrium was achieved. Then, the same amount of $AgNO_3$ was introduced into the electrolyte. The concentration of the Ag$^+$ ions in the electrolyte was tested after 7 h when the adsorption equilibrium was reached for the second time. The same procedure was performed for the third time to determine the amount of Ag$^+$ ions that the UiO-66 powders could adsorb.

### Electrochemical measurements

We determined the ionic conductivity of the pressed UiO-66 octahedron pellets and the electrolyte by electrochemical impedance spectroscopy (EIS). In detail, the UiO-66 octahedron powder was dried at 70 °C for 72 h to remove the solvents. Then, the powders were stored in the electrolyte for 3 h before centrifuging and drying for another 72 h at 70 °C. Finally, a certain amount of the collected UiO-66 octahedron powder was pressed into a pellet under a pressure of 3 MPa for 30 s. The UiO-66 pellet was placed between two stainless-steel electrodes in a CR2016 coin cell. 30 μl of the electrolyte solutions were introduced into the pellet and the redundant electrolyte was removed before the EIS tests. The electrolyte was filled into a customized Swagelok cell without the UiO-66 pellet to test its conductivity. An Autolab PGSTAT302N was used for the EIS measurements at 45 °C and the frequency ranged from $10^6$ to 0.1 Hz. The resistance of the samples was calculated based on the EIS curves. The ionic conductivity was obtained using Equation 4:[54]

$$\sigma = \frac{d}{SR_b} \qquad (4)$$

Where $S$ and $d$ represent the area and the thickness of the UiO-66 octahedron pellet, respectively. $R_b$ is the resistance and $\sigma$ is the ionic conductivity.

### Excess chemical potentials of Ag+ ions in the UiO-66 octahedra and the bulk solutions

We performed MD simulations using the GROMACS 2020.6 package[55]. The equations of motion were integrated with the leapfrog algorithm using a time step of 2.0 fs. Non-bonded potentials (Lennard−Jones (LJ) and Coulombic interactions) were truncated at 1.0 nm and were shifted by a constant such that they were zero at the cut-off ("Potential-shift" function in GROMACS). Long-range Coulombic interactions were

evaluated using the particle mesh Ewald method. The LJ interactions between unlike particles were represented by the Lorentz–Berthelot combination rules. The periodic boundary conditions were applied to all the directions. In the production runs, the temperature ($T$) and pressure ($p$) were controlled using Nosé–Hoover thermostat and Parrinello-Rahman barostat with a damping constant of 2 and 5 ps, respectively. In the equilibrations, the Berendsen algorithm with a damping constant of 1 ps for both $T$ and $p$ was used. The temperature was set to 318 K in all the simulations, which is the same as the temperature in our experiments. The unit cell structure of UiO-66 and the force fields for the components (Zr, C, O, and H) were taken from Yang's paper[56], which can reasonably reproduce the conductivity of anions in the UiO-66 supercell[33]. Ag and nitrate ions were represented by the force field developed by Mertz et al.[57]. SDS and water molecules were represented by the GAFF force field[58] and the TIP3P model[59], respectively. For each calculation below, the standard deviation was estimated from three independent simulations started from different initial configurations. We estimated the excess chemical potential of an Ag$^+$ ion into UiO-66 octahedra and into the bulk SDS aqueous solutions. The former system consisted the $4 \times 1 \times 1$ supercell of UiO-66 and 340 water molecules. The water loading of 15 mmol g$^{-1}$ is close to the experimental value of 16 mmol g$^{-1}$ at 298 K and 1 bar[60]. The UiO-66 supercell was fixed in the space and the MD simulations were conducted under the canonical (NVT) ensemble.

The latter system consisted of 22 SDS molecules and 8510 water molecules in a cubic box. The concentration is the same as that used in our experiments. A nitrate ion was also added as a counter-anion. The MD simulations were conducted under the isothermal-isobaric (NpT) ensemble at 1 bar.

The excess chemical potential ($\mu^{ex}_{Ag}$) of Ag$^+$ ion was computed through the Bennett acceptance ratio method[61] using 21 MD simulations from the fully coupled state to the fully decoupled state. The intermolecular interactions between the Ag$^+$ ion and the other molecules were decoupled in stepwise. The Coulombic interactions were turned off from the fully-coupled state (a factor of 1.0) to the fully-decoupled state (0.0) with an increment of 0.1 with maintaining the LJ interactions. Then, in the absence of Coulombic interactions, the LJ interactions were decoupled from a factor of 0.9 to 0.0 with an increment of 0.1. The LJ interactions were parameterized using a soft-core potential[62] to prevent "end-point catastrophes". At each condition, an initial configuration was first energy-minimized using the steepest descent method. Then, a 1 ns equilibration run was followed by a 20 ns production run.

### Diffusion behavior of Ag$^+$ ions within UiO-66 octahedra

We computed the diffusion behavior of Ag$^+$ ions in the UiO-66 framework under an electric field of 5.0 V nm$^{-1}$ along the **x**-axis. The system was prepared in the same manner as above, except the concentration of Ag$^+$ ions was different. The number of Ag$^+$ ions were varied from 10 to 80 with an interval of 10. At each condition, a 1 ns equilibration run was followed by a 100 ns production run. We used the rigid water molecule because the polarization of water can be neglected under this magnitude of the electric field[42].

### FDTD simulations

The commercial software Lumerical Solutions was used to perform the FDTD simulations. For the periodic array, a plane wave was employed at the normal incidence. The electromagnetic field distributions over the Ag nanotriangle array were performed as follows. A 2D-simulation region of 1.7 μm × 1.6 μm was used to cover one unit on the surface of the array. The model was imported from the SEM image (Fig. 2k). The PML boundary condition was set for the z direction, while the anti-symmetric and symmetric boundary conditions were used for the x and y directions, respectively. In all simulations, the mesh size in all directions (x, y, and z) was set as 2 nm, and the mesh accuracy was

increased to 5. The simulation time was 1000 fs. The optical constants of Ag were taken from Johnson and Christy at 532 nm, from 350 to 900 nm.

### SERS measurement

For the detection of R6G molecules, the Ag nanotriangle arrays were immersed in 1.5 ml of R6G solutions at different concentrations. The substrates were taken out after 2 h and dried in the ambient conditions. The SERS spectra were collected using a confocal microprobe Raman system (Renishaw In-via Reflex). The excitation laser wavelength was 532 nm with a power of ~ 0.2 mW. The laser beam was focused on the samples through a 50 × objective. The limit of detection of the R6G molecules using the Ag nanotriangle array was obtained with 2 s integration time and 5 acquisitions.

## Data availability

All experimental data within the article and its Supplementary Information are available from the corresponding authors upon request. Source data are provided with this paper.

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

## Acknowledgements

This work was supported by Key R&D Program of Zhejiang Province (2023C01088), National Natural Science Foundation of China

(52273233, 22250610195, and 22273083), Zhejiang Provincial Natural Science Foundation of China (LR19E010001), the National Key Research and Development Program of China (2018YFB0703803), and the Open Research Program of Key Laboratory of 3D Micro/Nano Fabrication and Characterization of Zhejiang Province, Westlake University. Part of the work was conducted in the ZJU micro-nanofabrication centre. We would like to thank Prof. Haobin Wu and Dr. Wei Zhao for their help in preparing the UiO-66@PBI composite films.

## Author contributions

S.Y. conceived the idea. S.Y., L.Z. and Y.L. designed the experiments. Y.L. carried out the material synthesis and characterizations. Xu.Z., H.L. and K.M. performed simulations. S.Y., Y.L., M.Y. and Xi. Z. analyzed the data and draw the figures. All authors contributed to the writing of the manuscript. All authors reviewed the manuscript.

## Competing interests

The authors declare no competing interests.
