## [Peer Review file · Nature Communications]

REVIEWER COMMENTS

Reviewer #1 (Remarks to the Author):

This is a very interesting, and unexpected, result that will appeal to a large audience. The nanostructures are quite beautiful. I really only have a few comments and suggestions for the authors,

1. When I first saw the title of the paper, I assumed that the nanostructures will be built based on the uniform pore sizes in the MOF. In fact they are determined by the size of the MOF octahedra that are produced. Hence, the processing of this octahedra is probably the limiting factor in this work. Some of these "nanostructures" are almost on the micrometer scale.

2. The use of "self-assembled monolayer" also had me expecting molecular level SAMS.

3. It is still hard for me to picture that the Ag(I) ions would prefer to be transported through the very small pore of the MOF instead of by the solution. Can the authors rule this out?

4. In the methods section, line 306, it was stated that the deposition voltage was maintained at -1.2 V. Was this a deposition voltage (or bias) or a potential versus some reference electrode? Was this a two or three electrode system?

5. There are a lot of grammatical issues. I will point out a few,

Line 102, "approprait" should be "appropriate."

Line 104, "invovled" should be "involved"

Line 146, what are "large junks"?

Line 168, "influed" should be "influenced"

Line 239, "formed one after" should be "formed after"

Line 310, "5.0 KV" should be "5.0 kV"

Reviewer #2 (Remarks to the Author):

This paper reports the employment of a monolayer of metal organic framework (MOF) octahedra as a template to guide patterned metal electrodeposition. Notably, metallic films were grown underneath

the MOF octahedral, yielding unusual metallic surface patterns. While this MOF-guided electrochemical lithography is distinctly different from the traditional colloidal lithography methods, the application range, adjustability, and accuracy of this method are limited. Particularly, the overall quality of the produced surface patterns is not high as they are generally made of aggregates of irregular particles, leading to coarse surfaces and edges. The templating process was investigated carefully but little efforts were devoted to exploring the properties and potential applications of the obtained metallic surface patterns. Considering the large number of reports on templating or lithography approaches toward well-defined surface patterns, the current work may not represent such a significant advance worthy publication in Nature Communications. This manuscript may be publishable elsewhere after appropriate revision.

Specific comments:

1. Some scale bars in Figure 2 are hard to discern, the scale bar in the right panel of Fig. 2g is incorrect. Furthermore, the exact preparation conditions for the samples shown in Fig. 2h and I are not known.
2. It was shown that SDS played important roles in refining the electrodeposited Ag grains. However, the roles played by the SDS remain largely unclear. It was claimed that SDS lowered the surface tension of the electrolyte, promoting the electrolyte to enter the nanochannels. This speculation is highly doubtful considering that SDS lowered the surface tension of the electrolyte based on the adsorption of the SDS molecules on the solution surface but the SDS molecules can not enter the nanochannels themselves. The authors may want to compare the chemical potentials of the silver ions in the SDS solution and inside the channels, which could shed some light on the difference in their chemical reactivity.
3. The MOF template-induced electrodeposition of complex Ag surface nanopatterns was investigated to show the recyclability of the MOF template. However, such a demonstration of applications seems trivial. It would be more persuasive if the authors could prepare high-quality surface patterns exhibiting novel plasmonic properties and/or promising applications.
4. There are some typos. For example, "Exploring ... are" (Abstract).

Reviewer #3 (Remarks to the Author):

In the study conducted by Yang et al. on "Metal Organic Framework Template-Guided Electrochemical Lithography", MOF octahedra assembled monolayer was employed as a template for metal electrodeposition of metal, revealing an unidentified guiding growth mode. They reported the growth of metallic films directly underneath the MOF octahedra. Through their experimental measurements and the molecular dynamics simulations, they showed the rapid ion transport within the MOF nanochannels accounted for the guiding growth mode.

Although the idea of self-assembled UiO-66 monolayer that is used in this study (see Cui et al. *Small*, 2014, 10, 3672, <https://doi.org/10.1002/sml.201302983>) and MOF-templated electrodeposition of metals (See *Electrochimica Acta*, 2016, 222, 361-369, <https://doi.org/10.1016/j.electacta.2016.10.187>) is not new, the concept of metal patterning based templated by self-assembled MOF superstructures is interesting and could benefit experts working in the diverse areas of materials science such as integrated circuits, optoelectronics and photocatalysis. However, I have some concerns that are related to the detailed structure of the deposited metal. The

authors have devoted more time to describing the self-assembly of UiO-66 and the ion diffusion process but giving less attention to the structure and property of the metal. Specifically, could the authors address the following?

1. How does the pore size and shape of UiO-66 octahedron cavity impact on the structure of the deposited metal? Should there be a control in the shape and /or morphology of the metal due to effect of the nanochannels of the MOF, in addition to the nanotriangles that are formed?
2. As a follow-up to question 1, if the MOF channels do not control the structure of the deposited metal, then why should the reader consider MOF-templating over colloidal templating?
3. Could this concept be demonstrated to deposit other metals, e.g., Au, Cu or Pd?
4. Could MOF-guided electrochemical lithography be demonstrated with another MOF superstructure (e.g., ZIF-8, see "Avci, et al. Nature Chem 2018, 78–84. <https://doi.org/10.1038/nchem.2875>")? If not, why not?
5. The articles by Cui et al. *Small*, 2014, 10, 3672, <https://doi.org/10.1002/sml.201302983> and Worrall et al., *Electrochimica Acta*, 2016, 222, 361-369, <https://doi.org/10.1016/j.electacta.2016.10.187> should be discussed in the introductory section.
6. In Figure 2h and I, elemental mapping analysis is recommended rather than false colouring.
7. TEM characterisation of the deposited metal is recommended.
8. The difference in the optical behaviour of the metal deposited with and without the MOF template should be discussed.

Overall, I do not recommend the manuscript to be accepted in the current version as I believe that the characterisations and discussions are incomplete.

Reviewer #4 (Remarks to the Author):

In their manuscript the authors report on the electrodeposition of periodic metallic nanostructures using a template process. The template was produced by assembling MOF-nanoparticles into self-supporting 2D layers or multilayers. The assembled particles exhibited a high degree of orientation, as evidenced by XRD data. The preparation of monodisperse UiO-66 nanoparticles, yielding octahedra, is well-established in the literature. The templates were then deposited on a solid substrate and used as the cathode in connection with a carbon-counterelectrode for the electrodeposition of Ag. Using SEM, the authors demonstrated that - depending on the electrodeposition conditions - well-defined Ag nanoparticles were formed below the UiO-66 octahedra. This phenomenon was explained by the channels within the (oriented) MOF nanoparticles acting as channels for the Ag ion transport.

I find the results rather interesting and would, in principle, recommend publication in Nature Communications.

Before the paper can be accepted, the authors should consider the following point.

- 1) The authors state: "The electrodeposition was carried out using the UiO-66 octahedron monolayer as the cathode and a carbon rod as the counter electrode." UiO 66 is an insulator - how can a

monolayer made from this material be used as a cathode? Please also compare to the recent paper on conductivity in UiO-66 pellets from Janek's group (DOI: 10.1002/batt.20220318).

2) The text is in part difficult to read. A native speaker of the English language should go through the manuscript.

3) It would be interesting to get some further information on the properties of the deposited Ag material, e.g. on its electrical conductivity.

After the authors have considered these points I expect the paper to be suitable for publication in Nat. Comm.

Responses to Referees' comment

Reviewer #1:

Overall Comment: *This is a very interesting, and unexpected, result that will appeal to a large audience. The nanostructures are quite beautiful. I really only have a few comments and suggestions for the authors.*

Author response: We greatly appreciate your positive and constructive comments, which have significantly improved the quality of our work. We have made revisions according to your valuable suggestions in the revised manuscript. Once again, thank you for your valuable input, and we look forward to your further insights.

Comment 1: *When I first saw the title of the paper, I assumed that the nanostructures will be built based on the uniform pore sizes in the MOF. In fact they are determined by the size of the MOF octahedra that are produced. Hence, the processing of this octahedra is probably the limiting factor in this work. Some of these "nanostructures" are almost on the micrometer scale.*

Response 1: Thank you for your valuable comments. Indeed, the MOF nanochannels are good templates to prepare nanostructures within them. For example, Prof. Terasaki, *et al.* filled the mesopores of MOF with TiO₂ using wet chemical methods (*Nature* 2020, 586, 549). Regarding our current work, the exclusive electrodeposition growth occurring at the MOF/substrate interface was also unexpected to us. To address any possible confusion between nanochannel templating and nanochannel guiding, we have revised the title to "Metal Organic Framework Template-Guided Electrochemical Lithography on Substrates for SERS Sensing Applications". This modification better captures the unique growth mode we have observed. The MOF-guided electrochemical growth mode opens possibilities in the template-based nanostructure fabrication fields.

As you said, the size of the electrodeposited structures is determined by the surface area of the MOF octahedra facets. Here, we chose to use microscale MOF octahedra because they were easy to assemble into a monolayer film. However, we also acknowledge that reducing the size of the MOF particles to a few tens of nanometers could result in smaller electrodeposited nanostructures. Additionally, we would like to reference the work of Prof. Rob Ameloot *et al.*, who demonstrated the etching of MOF films using X-ray and electron-beam lithography methods to design desired nanopatterns with nanometer-scale precision (*Nat. Mater.* 2021, 20, 93, **Fig. R1**). We expect to transform these MOF nanopatterns into metallic surface nanopatterns, and importantly, our methods allow for the repeated utilization of these MOF

nanopatterns. An even more exciting application of the MOF-guided electrochemical growth involves utilizing a single MOF nanoparticle anchored to an atomic force microscope tip capable of programmatically moving at the nanoscale to electrochemically “print” complex 3D metallic nanopatterns. This would enable the electrochemical “printing” of complex 3D metallic nanopatterns. We are currently in the process of constructing the necessary equipment for this endeavor.

Fig. 4 | High-resolution EBL patterning of 100 nm thick ZIF-71 films. a,b, SEM (left) and AFM (right) topographic images of the ZIF-71 film with 50 nm trenches (a) and 200 nm lines (b). **c,d** SEM images of the ZIF-71 film with 100 nm (c) and 60 nm (d) lines. **e,f**, SEM images of the ZIF-71 film with 40 nm holes (e) and a grid with a 30 nm line width (f). Scale bars: **a,b**, 1 μm ; **c**, 1 μm (200 nm for inset); **d**, 200 nm (100 nm for inset); **e,f**, 100 nm (50 nm for inset).

Fig. R1 Copied from *Nat. Mater.* 2021, 20, 93. The authors used electron beam lithography method to etch ZIF-71 films into various nanopatterns.

Furthermore, we have expanded the versatility of the MOF-guided method in the revised manuscript by exploring more metals and MOFs. We successfully demonstrated the preparation of Cu nanotriangles using the monolayer UiO-66 octahedron template (**Fig. R2**), as well as the fabrication of Ag nanopillars using the MIL-96 template. Notably, the Ag nanopillars could grow into several micrometers in height (**Fig. R2**). Considering the extensive variety of over 20,000 MOFs available, we anticipate that the MOF-guided lithography technique holds potentials for the electrodeposition of numerous metal nanostructures.

Fig. R2 Preparation of Ag and Cu nanotriangle arrays using UiO-66 template and the Ag nanopillar arrays using the MIL-96 template. a, Ag nanotriangle arrays prepared using the UiO-66 templates; **b,** Cu nanotriangle arrays obtained using the UiO-66 templates; **c,** Ag nanopillar arrays obtained using the MIL-96 templates.

Comment 2: *The use of "self-assembled monolayer" also had me expecting molecular level SAMS.*

Response 2: Thank you for your comments. To avoid possible misunderstanding, we have changed the “self-assembled monolayer” into “self-assembled monolayer MOF microparticle template” in the revised manuscript. We appreciate your attention to details and your contribution to improving the accuracy of our work.

Comment 3: *It is still hard for me to picture that the Ag(I) ions would prefer to be transported through the very small pore of the MOF instead of by the solution. Can the authors rule this out?*

Response 3: Thank you for your insightful comments. Above all, the UiO-66 octahedra formed a closed plane within the monolayer film. This arrangement effectively separates the solution above the top part of the octahedra from the bottom part (except for the defects within the monolayer octahedron template). As a result, the blocking plane at the middle part of the monolayer octahedron template prohibited the transport of silver ions via the solution route (**Fig. R3**). It’s worth noting that previous reports have highlighted the fast ion transportation within the nanochannels of MOF structures (for example, *Adv. Mater.*, 2018, 30, e1707476; *Nat. Mater.*, 2020, 19, 767-774.; *Adv. Energy Mater.*, 2022, 12, 2200501).

Fig. R3 Schematics of different sectional views of the monolayer UiO-66 octahedron film. Cut from the middle of the UiO-66 octahedra, from a random position of the gaps and from the middle of the gaps.

Moreover, we have provided experimental and theoretical evidence demonstrating the preferential adsorption of Ag^+ ions into UiO-66 nanochannels. We observed the shift of the adsorption equilibrium of the Ag^+ ions in the system by inductively coupled plasma-mass spectrometry (ICP-MS, **Fig. R4**). In our experiments, 100 mg of UiO-66 powders were immersed in a 1 ml of solutions containing of Ag^+ ions with a concentration of 0.3 M for 18 h. The concentration of Ag^+ ions over time was monitored by ICP-MS and compared to the concentration of the solution without UiO-66 powders. We observed a rapid decrease in the remaining concentration of Ag^+ ions in the solution initially and reached equilibrium after 7 hours, indicating adsorption equilibrium was achieved during this period (Note that due to the dilution process required for ICP-MS tests, the measured concentration without UiO-66 powders varied slightly, **Fig. R4a**). This was referred to as the 1st adsorption equilibrium. Subsequently, we introduced additional Ag^+ ions to the solution to disrupt the equilibrium, and the process was repeated to reach the 2nd and 3rd equilibrium states. The remaining concentrations of Ag^+ ions after adsorption equilibrium were shown in **Fig. R4b**. The decrease in the concentration of Ag^+ ions in the solution before and after adsorption confirmed the penetration of Ag^+ ions into the nanochannels of UiO-66. The gradual decrease in the difference of Ag^+ ions concentration before and after adsorption can be attributed to the gradual occupancy of the nanochannels by Ag^+ ions.

Fig. R4 Ag⁺ ion adsorption by UiO-66 octahedron powders. **a**, The concentration variation of the Ag⁺ ions remaining in the solution after adsorption by the UiO-66 powders for different times; **b**, The concentration of Ag⁺ ions remaining in the solution after introducing more Ag⁺ ions into the equilibrium adsorption system. 1st, 2nd, and 3rd represent the remaining concentration of Ag⁺ ions in the solution before and after UiO-66 adsorption.

The difference in chemical potential ($\Delta\mu$) determines the mass transfer direction, thus we calculated the $\Delta\mu$ of Ag⁺ ions between UiO-66 nanochannels and the bulk solution theoretically (**Fig. R5**). The chemical potential (μ) of Ag⁺ ions is decomposed into the ideal term and the excess term (μ^{ex}):

$$\mu = RT\ln\Lambda^3\rho + \mu^{ex} \quad (1)$$

Where k_B is the Boltzmann constant; Λ is the de Broglie thermal wavelength of Ag; ρ is the number density of Ag⁺ ions. The difference in chemical potential ($\Delta\mu$) was then illustrated as:

$$\Delta\mu = k_B T \ln \frac{\rho_{UiO-66}}{\rho_{sol}} + \mu_{UiO-66}^{ex} - \mu_{sol}^{ex} \quad (2)$$

We computed the μ_{UiO-66}^{ex} of -454.6 ± 2.3 kJ mol⁻¹ and μ_{sol}^{ex} of -429.7 ± 0.1 kJ mol⁻¹ through molecular dynamics (MD) simulations combined with the Bennett acceptance ratio method (see Method section for details). The difference of $\mu_{sol}^{ex} - \mu_{UiO-66}^{ex} = 24.9$ kJ mol⁻¹ indicates that Ag⁺ ions prefer to enter the nanochannels of UiO-66 until the ratio of $\frac{\rho_{UiO-66}}{\rho_{sol}}$ approaches approximately 1.2×10^4 , resulting in $\Delta\mu = 0$. However, it is important to consider the accommodation limit of Ag⁺ ions within UiO-66. Our MD simulations revealed that the population of Ag⁺ ions was highly concentrated at the eight tetrahedral cages and two octahedral cages in the unit cell of UiO-66 with a volume of 9.3 nm³, corresponding to the number density of ~ 1.1 nm⁻³. This information implies that when UiO-66 is immersed in an SDS solution of 0.38 nm⁻³, all the cages in the UiO-66 are occupied by Ag⁺ ions at the saturated density

ρ_{UiO-66}^{sat} of $\sim 1.1 \text{ nm}^{-3}$. We calculated the number density of Ag^+ ions from the adsorption equilibrium in **Fig. R4**, the saturated density, ρ_{UiO-66}^{sat} was $\sim 1.3 \text{ nm}^{-3}$, which is consistent with the predicted value by MD simulations. Therefore, both the experimental and the simulation results suggest that all the cages within UiO-66 are occupied by Ag^+ ions when it is equilibrated within the SDS solution used in our experiments.

Fig. R5 Schematic of the chemical potential of Ag^+ ions in the bulk solution and in the nanochannels of the UiO-66.

Comment 4: In the methods section, line 306, it was stated that the deposition voltage was maintained at -1.2 V . Was this a deposition voltage (or bias) or a potential versus some reference electrode? Was this a two or three electrode system?

Response 4: Thank you for your valuable comments. The MOF-guided electrodeposition process was not sensitive to the applied potentials or the electrodeposition mode used. The Ag nanotriangles were obtained over a broad potential range, both in the two-electrode and three-electrode systems, as shown in **Fig. R6** to **Fig. R8**. Constant current mode could also generate Ag nanotriangle arrays (**Fig. R8**). To simplify the experimental setup, we primarily employed a two-electrode system in this work, with a working electrode composing of a self-assembled monolayer UiO-66 octahedron film-coated Au substrate, and with a carbon rod serving as the counter electrode. We have included the above discussions on the electrodeposition mode and the potential information into the revised manuscript.

Fig. R6 SEM images of Ag nanotriangle arrays deposited using a two-electrode system using low voltages for a duration of 60 s. The concentrations of (C_{AgNO_3} , C_{SDS}) were set to (300 mM, 14 mM). The applied voltages for each image were as follows: **a**, 1.2 V; **b**, 1.0 V; **c**, 0.6 V.

Fig. R7 SEM images of Ag nanotriangle arrays deposited in a two-electrode system using high voltages for a duration of 5 s. The concentrations of (C_{AgNO_3} , C_{SDS}) were set to (300 mM, 14 mM). The applied voltages for each image were as follows: **a**, 1.2 V; **b**, 1.7 V; **c**, 2.4 V.

Fig. R8 SEM images of Ag nanotriangle arrays deposited in a three-electrode system using Hg/Hg₂SO₄ as the reference electrode. The concentrations of (C_{AgNO₃}, C_{SDS}) were set to (300 mM, 14 mM). The experimental conditions for each image were as follows: **a**, Applying a constant voltage of 0.6 V for a duration of 60 s; **b**, Applying a constant current density of 13.3 mA cm⁻² for a duration of 15 s; **c**, Applying a constant current density of 1.3 mA cm⁻² for a duration of 300 s.

Comment 5: *There are a lot of grammatical issues. I will point out a few,*

Line 102, "appropriaste" should be "appropriate."

Line 104, "invovled" should be "involved"

Line 146, what are "large junks"?

Line 168, "influeced" should be "influenced"

Line 239, "formed one after" should be "formed after"

Line 310, "5.0 KV" should be "5.0 kV"

Response 5: Thank you for your comments. We are sorry about these typos and we have corrected these typos in the revised manuscript. Thank you once again for your interest and positive comments to our work.

Reviewer #2 (Remarks to the Author):

Overall Comment: *This paper reports the employment of a monolayer of metal organic framework (MOF) octahedra as a template to guide patterned metal electrodeposition. Notably, metallic films were grown underneath the MOF octahedral, yielding unusual metallic surface patterns. While this MOF-guided electrochemical lithography is distinctly different from the traditional colloidal lithography methods, the application range, adjustability, and accuracy of this method are limited. Particularly, the overall quality of the produced surface patterns is not high as they are generally made of aggregates of irregular particles, leading to coarse surfaces and edges. The templating process was investigated carefully but little efforts were devoted to exploring the properties and potential applications of the obtained metallic surface patterns. Considering the large number of reports on templating or lithography approaches toward well-defined surface patterns, the current work may not represent such a significant advance worthy publication in Nature Communications. This manuscript may be publishable elsewhere after appropriate revision.*

Author response: We greatly appreciate your insightful comments and valuable suggestions, which have greatly improved the quality of our work. In traditional colloidal lithography methods, colloidal spheres typically act as masks or molds during the targeted material growth. However, we discovered that different from the solid polystyrene or silica beads in colloidal crystal templates, the MOF octahedra demonstrated the ability to guide the growth of the electrodeposits at the interface between the octahedra and the substrate (**Fig. R1**). This MOF-guided electrodeposition growth mode was unexpected and not straightforward. Therefore, we focused more on the study of the growth mechanism in the original manuscript. To strengthen this claim, we included additional experimental and simulation results in the revised manuscript to solidify that this unique template-guided growth mode arose from the fast ion diffusion within the MOF's nanochannels. Based on your suggestions, we further explored the structural tunability of the obtained surface nanopatterns. The Ag nanotriangles were formed by nanoparticles for short electrodeposition times, while became solid structures for long durations (**Fig. R2**). Moreover, we demonstrated the versatility of the MOF-guided electrochemical growth method by successfully extending it to other metals (such as, Cu) and various MOFs (including UiO-66, MIL-96, etc., **Fig. R2**). With more than 20,000 types of MOFs available, we anticipate that an extensive array of metal nanostructures can be electrodeposited using this approach.

According to your advice, we also investigated the plasmonic properties of the prepared Ag nanotriangle arrays and their applications as surface-enhanced Raman spectroscopy (SERS) sensors. This application example has been included in the revised manuscript (see Response 3).

We hope that our work will ignite renewed research interest in the seemingly mature field of colloidal lithography methods, by using the MOF particles as self-assembled templates for electrodeposition growth. We firmly believe that this MOF-guided electrochemical growth mode opens up vast possibilities for nanostructure fabrication. The more exciting application of the MOF-guided electrochemical growth involves using a single MOF nanoparticle fixed to, for example, an atomic force microscope tip capable of programmatically moving at the nanoscale, to electrochemically “print” 3D complex metallic nanopatterns. We are now constructing the instrument.

Fig. R1 Schematic of the masking and the molding growth mode of the conventional colloidal lithography and the discovered guiding growth mode of the MOF lithography.

Fig. R2 Illustration of various MOFs utilized in MOF lithography to prepare metal surface nanopatterns.

Comment 1: Some scale bars in Figure 2 are hard to discern, the scale bar in the right panel of Fig. 2g is incorrect. Furthermore, the exact preparation conditions for the samples shown in Fig. 2h and I are not known.

Response 1: Thank you for your valuable comments. We apologize for the incorrect and the invisible scale bars in Fig.2. In the revised manuscript, we ensured that all of the scale bars are clearly visible and accurately represented in all figures. Additionally, we have included comprehensive experimental parameters for each sample in the revised figure captions, both in the revised manuscript and the revised Supplementary Information. Thank you for bringing these concerns to our attention, and we appreciate your continued interest in our work.

Comment 2: It was shown that SDS played important roles in refining the electrodeposited Ag grains. However, the roles played by the SDS remain largely unclear. It was claimed that SDS lowered the surface tension of the electrolyte, promoting the electrolyte to enter the nanochannels. This speculation is highly doubtful considering that SDS lowered the surface tension of the electrolyte based on the adsorption of the SDS molecules on the solution surface but the SDS molecules cannot enter the nanochannels themselves. The authors may want to

compare the chemical potentials of the silver ions in the SDS solution and inside the channels, which could shed some light on the difference in their chemical reactivity.

Response 2: Thank you for your valuable advice. The monolayer UiO-66 octahedron film was hydrophobic with a contact angle of 122.8° (**Fig. R3**). Indeed, SDS could not enter the nanochannels of the UiO-66. However, introducing SDS into the silver nitrate aqueous solutions could significantly lower the surface tension of the electrolyte, facilitating wetting of the UiO-66 template by the electrolyte. This promotes better contact between the electrolyte and the UiO-66 template, enabling the electrodeposition process. Moreover, the electrochemical impedance spectroscopy (EIS) curve demonstrated the effects of SDS on the conductivity adjustment of the system. The presence of SDS reduced the conductivity of the solution, while increased the conductivity of the system comprising the UiO-66 pellet, attributed to the improved wetting between the electrolyte and the UiO-66 template (**Fig. R4**). These findings indicated that the inclusion of SDS in the silver nitrate aqueous solutions played a crucial role in facilitating wetting of the UiO-66 template and improving the conductivity of the system. We have incorporated these explanations and results into the revised manuscript, providing a clearer understanding of the role of SDS in our experiments. Thank you for your valuable input.

Fig. R3 a, b, The water contact angle of UiO-66 thin film and the gold substrate. c, Surface tension of different solutions. Black curve: solutions composed of different concentrations of SDS. Blue curve: solutions composed of different concentrations of AgNO_3 . Orange curve: solutions composed of SDS (14 mM, fixed) and different concentrations of AgNO_3 .

Fig. R4 Experimentally measured conductivities of the UiO-66 octahedron pellet and the electrolyte solutions. a, b, EIS plots of UiO-66 octahedron pellet and the electrolyte, respectively. **c,** The calculated conductivities of the UiO-66 pellet and the electrolyte. **d,** The increase and decrease of the conductivities of the UiO-66 pellet and the electrolyte after adding SDS.

According to your suggestions, we have provided experimental and theoretical evidence demonstrating the preferential adsorption of Ag^+ ions into the UiO-66 nanochannels. We observed the shift of the adsorption equilibrium of the Ag^+ ions in the system by inductively coupled plasma-mass spectrometry (ICP-MS, **Fig. R5**). In our experiment, 100 mg of UiO-66 powders were immersed in 1 ml of solutions containing of Ag^+ ions with a concentration of 0.3 M for 18 h. The concentration of Ag^+ ions over time was monitored by ICP-MS and compared to the concentration of the solution without UiO-66 powders. We observed a rapid decrease in the remaining concentration of Ag^+ ions in the solution initially and reached equilibrium after 7 hours, indicating adsorption equilibrium was achieved during this period (Note that due to the dilution process required for ICP-MS tests, the measured concentration without UiO-66 powders varied slightly, **Fig. R5a**). This was referred to as the 1st adsorption equilibrium. Subsequently, we introduced additional Ag^+ ions to the solution to disrupt the equilibrium, and the process was repeated to reach the 2nd and 3rd equilibrium states. The remaining concentrations of Ag^+ ions after adsorption equilibrium were shown in **Fig. R5b**. The decrease in the concentration of Ag^+ ions in the solution before and after adsorption confirmed the penetration of Ag^+ ions into the nanochannels of UiO-66. The gradual decrease in the difference of Ag^+ ions concentration before and after adsorption was attributed to the gradual occupancy

of the nanochannels by Ag^+ ions.

Fig. R5 Ag^+ ion adsorption by UiO-66 octahedron powders. **a**, The concentration variation of the Ag^+ ions remaining in the solution after adsorption by the UiO-66 powders for different times; **b**, The concentration of Ag^+ ions remaining in the solution after introducing more Ag^+ ions into the equilibrium adsorption system. 1st, 2nd, and 3rd represent the remaining concentration of Ag^+ ions in the solution before and after UiO-66 adsorption.

The difference in chemical potential ($\Delta\mu$) determined the mass transfer direction, thus we calculated the $\Delta\mu$ of Ag^+ ions between UiO-66 nanochannels and the bulk solution theoretically (**Fig. R6**). The chemical potential (μ) of Ag^+ ions was decomposed into the ideal term and the excess term (μ^{ex}):

$$\mu = RT\ln\Lambda^3\rho + \mu^{ex} \quad (1)$$

Where k_B is the Boltzmann constant; Λ is the de Broglie thermal wavelength of Ag ; ρ is the number density of Ag^+ ions. The difference in chemical potential ($\Delta\mu$) was then illustrated as:

$$\Delta\mu = k_B T \ln \frac{\rho_{\text{UiO-66}}}{\rho_{\text{sol}}} + \mu_{\text{UiO-66}}^{ex} - \mu_{\text{Sol}}^{ex} \quad (2)$$

We computed the $\mu_{\text{UiO-66}}^{ex}$ of $-454.6 \pm 2.3 \text{ kJ mol}^{-1}$ and μ_{Sol}^{ex} of $-429.7 \pm 0.1 \text{ kJ mol}^{-1}$ through molecular dynamics (MD) simulations combined with the Bennett acceptance ratio method (see Method section for details). The difference of $\mu_{\text{Sol}}^{ex} - \mu_{\text{UiO-66}}^{ex} = 24.9 \text{ kJ mol}^{-1}$ indicated that Ag^+ ions preferred to enter the nanochannels of UiO-66 until the ratio of $\frac{\rho_{\text{UiO-66}}}{\rho_{\text{sol}}}$ approaching approximately 1.2×10^4 , resulting in $\Delta\mu = 0$. However, it is important to consider the accommodation limit of Ag^+ ions within UiO-66. Our MD simulations revealed that the population of Ag^+ ions was highly concentrated at the eight tetrahedral cages and two octahedral cages in the unit cell of UiO-66 with a volume of 9.3 nm^3 , corresponding to the number density

of $\sim 1.1 \text{ nm}^{-3}$. This information implied that when UiO-66 was immersed in an SDS solution of 0.38 nm^{-3} (**Fig. R5**), all the cages in the UiO-66 were occupied by Ag^+ ions at the saturated density $\rho_{\text{UiO-66}}^{\text{sat}}$ of $\sim 1.1 \text{ nm}^{-3}$. We calculated the number density of Ag^+ ions from the adsorption equilibrium in **Fig. R5**, the saturated density $\rho_{\text{UiO-66}}^{\text{sat}}$ was $\sim 1.3 \text{ nm}^{-3}$, which was consistent with the MD simulations predicted value. Therefore, both the experimental and the simulation results suggested that all the cages within UiO-66 were occupied by Ag^+ ions when it was equilibrated within the SDS solution used in our experiments.

Fig. R6 Schematic of the chemical potential of Ag ions in the bulk solution and in the nanochannels of the UiO-66.

Comment 3: *The MOF template-induced electrodeposition of complex Ag surface nanopatterns was investigated to show the recyclability of the MOF template. However, such a demonstration of applications seems trivial. It would be more persuasive if the authors could prepare high-quality surface patterns exhibiting novel plasmonic properties and/or promising applications.*

Response 3: Thank you for your valuable suggestions. We demonstrated recyclability of the MOF template considering that the relatively time-consuming fabrication process. Prof. Rob Ameloot *et al.* reported on the etching of MOF films using X-ray and electron-beam lithography methods to design desired nanopatterns with nanometer-scale precision (*Nat. Mater.* 2021, 20, 93, **Fig. R7**). This provides a potential avenue for creating MOF nanopatterns composed of fine structures using the time-consuming and expensive electron beam lithography technique. Considering the future use of such MOF nanopatterns, it is crucial to ensuring the recyclability of the MOF template.

Fig. 4 | High-resolution EBL patterning of 100 nm thick ZIF-71 films. **a,b**, SEM (left) and AFM (right) topographic images of the ZIF-71 film with 50 nm trenches (**a**) and 200 nm lines (**b**). **c,d** SEM images of the ZIF-71 film with 100 nm (**c**) and 60 nm (**d**) lines. **e,f**, SEM images of the ZIF-71 film with 40 nm holes (**e**) and a grid with a 30 nm line width (**f**). Scale bars: **a,b**, 1 μm ; **c**, 1 μm (200 nm for inset); **d**, 200 nm (100 nm for inset); **e,f**, 100 nm (50 nm for inset).

Fig. R7 Copied from *Nat. Mater.*, 2021, 20, 93. The authors used electron beam lithography method to etch ZIF-71 films into various nanopatterns.

According to your valuable suggestions, we also investigated the optical properties of the Ag nanotriangle arrays and their potential applications as surface-enhanced Raman spectroscopy (SERS) sensing substrates. We performed the finite-difference time-domain (FDTD) simulations to analyze the plasmonic properties of the Ag nanotriangle arrays (**Fig. R8**). The simulation results revealed four absorption peaks at approximately 450 nm, 530 nm, 550 nm, and 750 nm. The 450 nm and the 550 nm localized surface plasmon (LSPR) peaks were attributed to the quadrupolar and the dipolar mode of individual Ag nanotriangles, while the 530 nm and the 750 nm peaks arose from the plasmonic coupling between neighboring nanotriangles in the array (*J. Phys. Chem. C*, 2011, 115, 9291-9305; *J. Phys. Chem. C*, 2012, 116, 14591-14598). Experimental measurements of the absorption spectra exhibited two broad peaks centered at around 550 nm and 740 nm. Discrepancies between the FDTD simulated and the experimentally measured plasmonic properties were induced by the structural deviation from the perfect triangular shapes and the defects in the ordered array.

Furthermore, the FDTD simulations indicated strong electromagnetic fields located at the edges and the tips of the Ag nanotriangles (**Fig. R9**). These regions with enhanced electromagnetic fields could serve as SERS hot spots to significantly enhance the Raman signals of chemical molecules (*Science*, 2008, 321, 388-392; *Chem. Soc. Rev.*, 2019, 48, 731-756. *Nat. Rev. Phys.*, 2020, 2, 253-271). We used Rhodamine 6G (R6G) as a model molecule to evaluate the SERS sensing performance of the silver nanotriangle arrays. Clear SERS signals of R6G molecules even at a concentration of 10 nM were observed (**Fig. R10**). The relative

standard deviation (RSD) of the 612 cm^{-1} SERS peak intensity was estimated to be 6.37%, indicating good signal reproducibility.

Fig. R8 The simulated and measured extinction spectra of different structures. Dotted curve: FDTD simulated extinction spectrum of Ag nanotriangle array. Orange curve: Ag nanotriangle array after subtracting the influence of the gold substrate. Grey curve: Large Ag blocks electrodeposited on the gold substrate without the UiO-66 template. Black curve: Monolayer UiO-66 octahedron template.

Fig. R9 FDTD simulated electromagnetic field distribution over the Ag nanotriangle array. **a**, Schematic of the Ag nanotriangle array; **b**, Electromagnetic field distribution over the Ag nanotriangle array excited by a 532 nm laser.

Fig. R10 SERS performance of the Ag nanotriangle arrays. **a**, SERS spectra of R6G molecules at different concentrations on the Ag nanotriangle arrays; **b**, The intensity of the 612 cm^{-1} SERS peak at randomly chosen 36 sites on the Ag nanotriangle arrays.

Comment 4: *There are some typos. For example, “Exploring ... are” (Abstract).*

Response 4: Thank you for pointing out these typos. We’ve thoroughly reviewed the manuscript and corrected any typos or grammatical mistakes in the revised manuscript. Thank you again for your valuable suggestions.

Reviewer #3 (Remarks to the Author):

Overall Comment: *In the study conducted by Yang et al. on “Metal Organic Framework Template-Guided Electrochemical Lithography”, MOF octahedra assembled monolayer was employed as a template for metal electrodeposition of metal, revealing an unidentified guiding growth mode. They reported the growth of metallic films directly underneath the MOF octahedra. Through their experimental measurements and the molecular dynamics simulations, they showed the rapid ion transport within the MOF nanochannels accounted for the guiding growth mode. Although the idea of self-assembled UiO-66 monolayer that is used in this study (see Cui et al. Small, 2014, 10, 3672, <https://doi.org/10.1002/sml.201302983>) and MOF-templated electrodeposition of metals (See Electrochimica Acta, 2016, 222, 361-369, <https://doi.org/10.1016/j.electacta.2016.10.187>) is not new, the concept of metal patterning based templated by self-assembled MOF superstructures is interesting and could benefit experts working in the diverse areas of materials science such as integrated circuits, optoelectronics and photocatalysis. However, I have some concerns that are related to the detailed structure of the deposited metal. The authors have devoted more time to describing the self-assembly of UiO-66 and the ion diffusion process but giving less attention to the structure and property of the metal. Overall, I do not recommend the manuscript to be accepted in the current version as I believe that the characterisations and discussions are incomplete.*

Author response: Thank you for your valuable and constructive comments. We have cited the important work by Cui *et al.* regarding the self-assembly of MOF microparticles at the air/liquid interface. Also, we have cited the *Electrochimica Acta* paper reporting the electrodeposition of metals within the MOF nanochannels forming nanoclusters within the MOF microparticles. We have discussed the differences between our work and the two important papers in the revised manuscript. We greatly appreciate your interest to the MOF-templated metal array patterning concept and its potential applications in various areas of materials science. Since the MOF-guided growth mode was completely different from the masking and the molding mode in conventional colloidal lithography, we focused on the mechanism study of the MOF-guided growth mode in our original submission. In the revised manuscript, according to your suggestion, we further characterized the Ag nanotriangles using atomic force microscopy, providing additional insights into their structural properties. We have extended the MOF-guided lithography to various metals, such as Ag and Cu, and used different MOFs including UiO-66, ZIF-8, and MIL-96 to realize the MOF-guided growth. These results demonstrated the versatility and applicability of the MOF-guided growth mode. Additionally, we calculated the

difference in the chemical potentials of silver ions between the bulk solution and the nanochannels of UiO-66, explaining the reason why silver ions preferred to enter the UiO-66 octahedra via the nanochannels.

Comment 1: *How does the pore size and shape of UiO-66 octahedron cavity impact on the structure of the deposited metal? Should there be a control in the shape and /or morphology of the metal due to effect of the nanochannels of the MOF, in addition to the nanotriangles that are formed?*

Response 1: Thank you for your comments. The silver ions preferred to pass through the nanochannels within UiO-66. We have conducted adsorption experiments and performed chemical potential calculations to confirm this phenomenon. The details of these findings have been included in Fig.3 in the revised manuscript. Once the silver ions entered the nanochannels, they were transported to the bottom cathode electrode and subsequently reduced into silver atoms at the interface between the UiO-66 octahedron and the cathode electrode. This continuous growth process leads to the formation of silver nanotriangles (**Fig. R1**). We therefore named it the MOF-guided growth mode. This MOF-guided growth mode is completely different from the conventional colloidal lithography where the metals were grown at the interstitials of the monolayer microparticle template, instead of exactly underneath the template.

Fig. R1 Schematic of the masking and the molding growth mode of the conventional colloidal lithography and the discovered MOF-guided growth mode of the MOF lithography.

Furthermore, the shape of the electrodeposited silver nanostructure was determined by the interface shape of the MOF microparticle and the underneath substrate. We confirmed this relationship by preparing UiO-66 octahedra templates and MIL-96 truncated hexagonal bipyramid templates. The UiO-66 templates yielded silver nanotriangle arrays due to their triangular facets contacting with the substrate. While in the case of MIL-96, electrodeposition generated Ag nanostructures with a trapezoidal shape, mirroring the interface shape between MIL-96 and the substrate (**Fig. R2**). Considering that more than 20,000 types of MOFs have been developed so far, we anticipated to prepare various metallic surface nanopatterns using different MOFs.

Fig. R2 The relevance between the facet shape of the MOF particles and the shape of the electrodeposited metal nanostructure arrays. a, UiO-66 octahedra and its triangular facets; **b,** MIL-96 truncated hexagonal bipyramid and its trapezoidal facet. The dotted polygons marked the facet and the shape of the electrodeposited silver micropillar.

Comment 2: *As a follow-up to question 1, if the MOF channels do not control the structure of the deposited metal, then why should the reader consider MOF-templating over colloidal templating?*

Response 2: Thank you for your valuable comments. The nanochannels within the MOF play a critical role in directing metal ions to the interface of the MOF microparticle and the underneath substrate. These metal ions would be reduced at the interface area between the MOF microparticle and the substrate, leading to the formation of metal nanostructures with the shape the same as the interface of the MOF microparticle and the substrate. This MOF-guiding growth mode is completely different from the conventional colloidal lithography where the metals were grown at the interstitials of the monolayer microparticle template (**Fig. R1**). The thickness and the shape of the metallic nanostructures were determined by the electrodeposition time and the shape of the MOF microparticles (**Fig. R3**).

Fig. R3 Morphology and thickness evolution of the Ag nanotriangles as electrodeposition proceeded. a-e, 5 s, 15 s, 30 s, 60 s, and 300 s, respectively. Electrodeposition was performed in the electrolyte composed of 300 mM AgNO₃ and 14 mM SDS.

The MOF-guiding growth mode opened many possibilities in the metallic nanopatterning fields. Prof. Rob Ameloot *et al.* reported that the MOF film could be etched using the X-ray and electron-beam lithography methods to design desired nanopatterns with nanometer scale precision (*Nat. Mater.* 2021, 20, 93, **Fig. R4**). We expected to transform these MOF nanopatterns into metallic surface nanopatterns and more importantly, these MOF nanopatterns could be repeatedly used using our methods. The more exciting application of the MOF-guided electrochemical growth would be using a single MOF nanoparticle anchored onto, for example, an atomic force microscope tip capable of moving programmatically at nanoscale, to electrochemically “print” complex 3D metallic nanopatterns. We are now constructing the equipment.

Fig. 4 | High-resolution EBL patterning of 100 nm thick ZIF-71 films. a, b, SEM (left) and AFM (right) topographic images of the ZIF-71 film with 50 nm trenches (a) and 200 nm lines (b). c, d SEM images of the ZIF-71 film with 100 nm (c) and 60 nm (d) lines. e, f, SEM images of the ZIF-71 film with 40 nm holes (e) and a grid with a 30 nm line width (f). Scale bars: a, b, 1 μm; c, 1 μm (200 nm for inset); d, 200 nm (100 nm for inset); e, f, 100 nm (50 nm for inset).

Fig. R4 Copied from *Nat. Mater.* 2021, 20, 93. The authors used electron beam lithography method to etch ZIF-71 films into various nanopatterns.

Comment 3: *Could this concept be demonstrated to deposit other metals, e.g., Au, Cu or Pd?*

Response 3: Thank you for your comments. As an example, we successfully fabricated Cu nanotriangles utilizing the UiO-66 octahedron monolayer as the template, as shown in **Fig. R5**. The electrolyte employed in the electrodeposition process consisted of 1.2 M $\text{Cu}(\text{NO}_3)_2 \cdot 3\text{H}_2\text{O}$ and 14 mM SDS, with the pH of the electrolyte adjusted to 1 by using HNO_3 . The electrodeposition voltage was 1.8 V and the electrodeposition time was 180 s.

Fig. R5 SEM images and element mapping of Cu nanotriangle arrays using the UiO-66 templates. a, b, SEM images; c, Element mapping of Cu.

Comment 4: *Could MOF-guided electrochemical lithography be demonstrated with another MOF superstructure (e.g., ZIF-8, see “Avci, et al. Nature Chem 2018, 78–84. <https://doi.org/10.1038/nchem.2875>”)? If not, why not?*

Response 4: Thank you for your valuable suggestions. The *Nature Chemistry* paper assembled different MOF particles into three-dimensional superstructures showing vivid colors. This paper inspired us to perform this work and we have cited it in our manuscript. Following your recommendations, we attempted to electrodeposit silver nanostructures using different MOF microparticles. Initially, we synthesized ZIF-8 cubes and obtained a 2D ZIF-8 microcube monolayer through the self-assembly process at the air/liquid interface (**Fig. R6**). However, the ZIF-8 microcubes were dissolved slightly during the assembly process. Consequently, when using the ZIF-8 microcube monolayer as the template to perform electrodeposition of silver, we failed to obtain silver microcube arrays (**Fig. R7**). Instead, the silver was electrodeposited both underneath and around the ZIF-8 microcubes, and the ZIF-8 microcubes were almost completely dissolved after electrodeposition (**Fig. R7**). The instability of ZIF-8 microcubes in an aqueous environment, especially when the mass of water exceeds their own mass, has been previously reported (*Microporous Mesoporous Mater.*, 2019, 279, 201-210 and *Microporous*

Mesoporous Mater., 2019, 288, 109568). Moreover, the acidic nature of the electrolyte accelerated the disintegration process (*Chem. Mater.*, 2016, 28, 6960-6967). In summary, ZIF-8 microcubes were not stable for the electrodeposition of metal nanostructure arrays.

Instead, we successfully fabricated silver trapezoidal arrays using the monolayer MIL-96 truncated hexagonal bipyramid templates (**Fig. R8**). MIL-96 is a porous aluminum trimesate MOF comprising aluminum clusters and 1,3,5-benzenetricarboxylic acid. The MIL-96 structure consists of micropores with a diameter of 0.88 nm and an estimated pore-opening diameter of approximately 0.25 nm (*J. Am. Chem. Soc.*, 2006, 128, 10223-10230). The silver micropillars grew underneath the MIL-96 microparticles with the length even above several micrometers. Considering that more than 20,000 types of MOFs have been invented so far, we anticipated the ability to prepare various metallic surface nanopatterns using different MOFs.

Fig. R6 Synthesis and self-assembly of ZIF-8 microcubes. **a**, XRD of the synthesized ZIF-8 microcubes; **b**, SEM image of the ZIF-8 microcubes. **c**, **d**, Photo and SEM image of self-assembled monolayer ZIF-8 microcube film.

Fig. R7 Electrodeposition of silver using the monolayer ZIF-8 microcube templates. a, Silver structure electrodeposited using ZIF-8 templates; **b,** ZIF-8 templates after immersing in the electrolyte [(C_{Ag}NO₃, C_{SDS}) = (300 mM, 14mM)]; **c,** The pH value of the electrolyte.

Fig. R8 Characterization of the monolayer MIL-96 microparticle template and the electrodeposited silver micropillar arrays. a, SEM image of the MIL-96 microparticle template; **b,** XRD of the synthesized MIL-96 truncated hexagonal bipyramids; **c, d,** SEM images of the electrodeposited silver micropillar arrays; **e, f,** Element mapping of Ag and Al.

Comment 5: *The articles by Cui et al. Small, 2014, 10, 3672, <https://doi.org/10.1002/sml.201302983> and Worrall et al., Electrochimica Acta, 2016, 222, 361-369, <https://doi.org/10.1016/j.electacta.2016.10.187> should be discussed in the introductory section.*

Response 5: Thank you for your comments. We have compared our work with the two papers you mentioned in the introduction part of the revised manuscript. Regarding the assembly of the MOF microparticles, we used similar methods to the approach described in the *Small* paper with some improvements: 1) our method did not need the PVP functionalization process of the

UiO-66 before assembly; 2) we dropped the UiO-66 dispersions onto a hydrophilic glass slide, instead of using a tilted glass against the rim. These modifications resulted in an improved order of the self-assembled monolayer UiO-66 octahedron template. Additionally, we extended this assembly method to various MOFs of different shapes and sizes (**Fig. R9**).

In the *Electrochimica Acta* paper, the authors used the nanochannels within the MOF particles as nanoscale templates for the electrodeposition of gold. They prepared gold nanoparticles or nanowires within the nanochannels of the MOF microparticles. In our work, we used the nanochannels within UiO-66 as transportation pathway for silver ions. These silver ions reached the interface between UiO-66 and the substrate, giving rise to the formation of silver nanotriangles on the substrate. We have included the above discussions in the introduction part of the revised manuscript.

Fig. R9 The self-assembly of various MOFs into a monolayer microparticle film. **a**, UiO-66; **b**, ZIF-8; **c**, MIL-96.

Comment 6: *In Figure 2h and I, elemental mapping analysis is recommended rather than false colouring.*

Response 6: Thank you for your valuable suggestions. We agree that incorporating elemental mapping analysis can enhance the clarity of the characterizations. Therefore, we included it in the revised manuscript to provide a clear spatial distribution of different elements (**Fig. R10**).

Fig. R10 Element mapping of Ag nanotriangles array. **a**, Top view; **b**, Side view; **c**, Side view with the UiO-66 template.

Comment 7: TEM characterisation of the deposited metal is recommended.

Response 7: Thank you for your valuable suggestions. The silver nanotriangles were grown on the silicon wafer. It is difficult to cut the thick silicon substrate into < 100 nm to perform TEM characterizations. We tried to peel off the silver nanotriangles from the substrate by ultrasonic treatment in water. However, these silver nanotriangles were tightly adhered to the substrate. Therefore, we used atomic force microscopy (AFM) characterization to characterize the surface roughness and the thickness of the silver nanotriangles (**Fig. R11**).

Fig. R11 AFM images of Ag nanotriangle arrays obtained at different electrodeposition times.

Comment 8: The difference in the optical behaviour of the metal deposited with and without the MOF template should be discussed.

Response 8: Thank you for your valuable suggestions. The Ag nanotriangle array electrodeposited using the monolayer UiO-66 octahedron templates exhibited bright structural color, while large Ag blocks formed with nanodendrites were formed without the UiO-66 template (**Fig. R12**). We performed the finite-difference time-domain (FDTD) simulations to analyze the plasmonic properties of the silver nanotriangle arrays (Error! Reference source not found.). The simulation results revealed four absorption peaks at approximately 450 nm, 530 nm, 550 nm, and 750 nm. The 450 nm and the 550 nm localized surface plasmon (LSPR) peaks are attributed to the quadrupolar and the dipolar mode of individual silver nanotriangles, while the 530 nm and the 750 nm peaks arose from the plasmonic coupling between neighboring nanotriangles in the array (*J. Phys. Chem. C*, 2011, 115, 9291-9305; *J. Phys. Chem. C*, 2012, 116, 14591-14598). Experimental measurements of the absorption spectra exhibited two broad peaks centered around 550 nm and 740 nm. Discrepancies between the FDTD simulated and the experimentally measured plasmonic properties were induced by structural deviation from the perfect triangular shapes and the defects in the ordered array. The extinction spectrum of the large Ag blocks on the gold substrate was almost the same as that of the gold substrate, because the large blocks only scattered on the substrate in a low number density.

Furthermore, the FDTD simulations indicated strong electromagnetic fields located at the edges and the tips of the Ag nanotriangles (**Fig. R9**). These regions with enhanced electromagnetic fields could serve as SERS hot spots to significantly enhance the Raman signals of chemical molecules (*Science*, 2008, 321, 388-392; *Chem. Soc. Rev.*, 2019, 48, 731-756. *Nat. Rev. Phys.*, 2020, 2, 253-271). We used Rhodamine 6G (R6G) as a model molecule to evaluate the SERS sensing performance of the silver nanotriangle arrays. Clear SERS signals of R6G molecules even at a concentration of 10 nM were observed (**Fig. R10**). The relative standard deviation (RSD) of the 612 cm^{-1} SERS peak intensity was estimated to be 6.37%, indicating good signal reproducibility.

Fig. R12 The comparison of electrodeposited Ag structures with and without UiO-66 octahedron templates. **a**, the SEM image and photo of Ag nanotriangles array electrodeposited with UiO-66 octahedron film; **b**, the SEM image and photo of Ag blocks electrodeposited without UiO-66 octahedron film. Electrodeposition was performed in the electrolyte composed of 300 mM AgNO₃ and 14 mM SDS. The applied voltage was 1.2V with a duration of 15 s.

Fig. R13 The simulated and measured extinction spectra of different structures. Dotted curve: FDTD simulated extinction spectrum of Ag nanotriangle array. Orange curve: Ag nanotriangle array after subtracting the influence of the gold substrate. Grey curve: Large Ag blocks electrodeposited on the gold substrate without the UiO-66 template. Black curve: Monolayer UiO-66 octahedron template.

Fig. R14 FDTD simulated electromagnetic field distribution over the Ag nanotriangle array. **a**, Schematic of the Ag nanotriangle array; **b**, Electromagnetic field distribution over the Ag nanotriangle array excited by a 532 nm laser.

Fig. R13 SERS performance of the Ag nanotriangle arrays. **a**, SERS spectra of R6G molecules at different concentrations on the Ag nanotriangle arrays; **b**, The intensity of the 612 cm^{-1} SERS peak at randomly chosen 36 sites on the Ag nanotriangle arrays.

Reviewer #4 (Remarks to the Author):

Overall Comment: *In their manuscript the authors report on the electrodeposition of periodic metallic nanostructures using a template process. The template was produced by assembling MOF-nanoparticles into self-supporting 2D layers or multilayers. The assembled particles exhibited a high degree of orientation, as evidenced by XRD data. The preparation of monodisperse UiO-66 nanoparticles, yielding octahedra, is well-established in the literature. The templates were then deposited on a solid substrate and used as the cathode in connection with a carbon-counter electrode for the electrodeposition of Ag. Using SEM, the authors demonstrated that - depending on the electrodeposition conditions - well-defined Ag nanoparticles were formed below the UiO-66 octahedra. This phenomenon was explained by the channels within the (oriented) MOF nanoparticles acting as channels for the Ag ion transport. I find the results rather interesting and would, in principle, recommend publication in Nature Communications.*

Author response: We greatly appreciate your interest in our manuscript and positive comments. Your insightful comments have further improved the quality of our manuscript. We hope that our revised manuscript now meets the high standards expected for publication in Nature Communications.

Comment 1: *The authors state: "The electrodeposition was carried out using the UiO-66 octahedron monolayer as the cathode and a carbon rod as the counter electrode." UiO-66 is an insulator - how can a monolayer made from this material be used as a cathode? Please also compare to the recent paper on conductivity in UiO-66 pellets from Janek's group (DOI: 10.1002/batt.20220318).*

Response 1: Thank you for your valuable comments. We regret that our original manuscript was apparently insufficiently clear and caused a misunderstanding. The UiO-66 octahedron monolayer floating on the water surface was picked up by a piece of gold-covered silicon wafer and used as the cathode electrode. The silver ions could enter the nanochannels of the UiO-66 octahedra and reach the underneath conducting gold substrate. These silver ions were reduced into silver atoms at the interface of the UiO-66 octahedra and the gold surface, giving rise to the formation of the silver nanotriangles. We performed further calculations about the chemical potentials of the silver ions in the bulk solution and the nanochannels of the UiO-66, further consolidating that the silver ions preferred to enter the nanochannels. The UiO-66 octahedron monolayer on the gold substrate was only slightly influenced the conductivity. According to

your suggestions, we tried to evaluate the conductivity of the UiO-66 octahedra formed pellet by the electrochemical impedance spectroscopy (EIS) measurements. The conductivity of the UiO-66 octahedron pellet was of the same order of magnitude as that of the electrolyte ($\sim 10^{-3}$ - 10^{-2} S cm^{-1} , **Fig. R1**). Notably, our focus was not on evaluating the absolute value of the UiO-66 pellet's conductivity (which needs careful design and fabrication process in the field of quasi-solid-state battery), but rather on comparing the conductivity variation of the UiO-66 pellet and the electrolyte after the addition of SDS. We performed the conductivity measurements of the UiO-66 pellets in a semi-humid environment rather than in a solid state. Specifically, we introduced 30 μl of the electrolyte onto the pellet (with a little amount of excess electrolyte, therefore could significantly increase the conductivity of the UiO-66 pallet) and kept the cell at 45 $^{\circ}\text{C}$ for 1 hour prior to conducting the EIS tests (**Fig. R2**). The conductivity of the UiO-66 octahedron pallet was improved after introducing SDS, while the conductivity of the bulk solution was reduced after introducing SDS. Prof. Janek, *et al.* utilized porous UiO-66 as an absorbent to hold Mg-ionogel electrolyte (MgIL) and assessed the conductivity of UiO66-MgIL. The resulting UiO66-MgIL formed a reversible quasi-solid-state magnesium battery with a typical conductivity range of $\sim 10^{-3}\sim 10^{-5}$ S cm^{-1} . This publication was a good demonstration to use UiO-66 in the battery field, which has been cited in our revised manuscript.

Fig. R1 Experimentally measured conductivities of the UiO-66 octahedron pellet and the electrolyte solutions. a, b, EIS plots of UiO-66 octahedron pellet and the electrolyte, respectively. **c,** The calculated conductivities of UiO-66 pellet and electrolytes. **d,** The increase and decrease of the conductivities of the UiO-66 pellet and the electrolyte after adding SDS.

Fig. R2 Equipment used for the electrochemical measurements. a, b, The customized swagelok cell. The diameter and the thickness of the chamber were 0.73 cm and 0.51 cm, respectively. **c, d,** The pressed UiO-66 octahedron pellet for EIS measurements. The diameter and the thickness of the pressed pellet were 0.65 cm and 0.03 cm, respectively.

Comment 2: *The text is in part difficult to read. A native speaker of the English language should go through the manuscript.*

Response 2: Thank you for your comments. We apologize for any difficulties you experienced while reading it. We have asked an English speaker to thoroughly review and edit our manuscript.

Comment 3: *It would be interesting to get some further information on the properties of the deposited Ag material, e.g. on its electrical conductivity.*

Response 3: Thank you for your suggestions. According to your valuable suggestions, we also investigated the optical properties of the silver nanotriangle arrays and their potential applications as surface-enhanced Raman spectroscopy (SERS) sensing substrates. We performed the finite-difference time-domain (FDTD) simulations to analyze the plasmonic properties of the silver nanotriangle arrays (Error! Reference source not found.). The simulation results revealed four absorption peaks at approximately 450 nm, 530 nm, 550 nm, and 750 nm. The 450 nm and the 550 nm localized surface plasmon (LSPR) peaks are attributed to the quadrupolar and the dipolar mode of individual silver nanotriangles, while the 530 nm and the 750 nm peaks arose from the plasmonic coupling between neighboring nanotriangles in the

array (*J. Phys. Chem. C*, 2011, 115, 9291-9305; *J. Phys. Chem. C*, 2012, 116, 14591-14598). Experimental measurements of the absorption spectra exhibited two broad peaks centered around 550 nm and 740 nm. Discrepancies between the FDTD simulated and the experimentally measured plasmonic properties were induced by structural deviation from the perfect triangular shapes and the defects in the ordered array.

Furthermore, the FDTD simulations indicated strong electromagnetic fields located at the edges and the tips of the silver nanotriangles (Error! Reference source not found.). These regions with enhanced electromagnetic fields could serve as SERS hot spots to significantly enhance the Raman signals of chemical molecules (*Science*, 2008, 321, 388-392; *Chem. Soc. Rev.*, 2019, 48, 731-756. *Nat. Rev. Phys.*, 2020, 2, 253-271). We used Rhodamine 6G (R6G) as a model molecule to evaluate the SERS sensing performance of the silver nanotriangle arrays. Clear SERS signals of R6G molecules even at a concentration of 10 nM were observed (Error! Reference source not found.). The relative standard deviation (RSD) of the 612 cm^{-1} SERS peak intensity was estimated to be 6.37%, indicating good signal reproducibility.

Fig. R13 The simulated and measured extinction spectra of different structures. Dotted curve: FDTD simulated extinction spectrum of Ag nanotriangle array. Orange curve: Ag nanotriangle array after subtracting the influence of the gold substrate. Grey curve: Large Ag blocks electrodeposited on the gold substrate without the UiO-66 template. Black curve: Monolayer UiO-66 octahedron template.

Fig. R14 FDTD simulated electromagnetic field distribution over the Ag nanotriangle array. **a**, Schematic of the Ag nanotriangle array; **b**, Electromagnetic field distribution over the Ag nanotriangle array excited by a 532 nm laser.

Fig. R3 SERS performance of the Ag nanotriangle arrays. **a**, SERS spectra of R6G molecules at different concentrations on the Ag nanotriangle arrays; **b**, The intensity of the 612 cm^{-1} SERS peak at randomly chosen 36 sites on the Ag nanotriangle arrays.

REVIEWERS' COMMENTS

Reviewer #1 (Remarks to the Author):

The authors have greatly improved the paper by carefully addressing comments by all of the the reviewers. I am still confused as to why metal ions would prefer to diffuse through the MOF instead of direct reduction at the metal surface between the MOF particles. I realize that transport of metal ions through the MOF can be relatively fast, but do they really transport faster in the pores than in the solution? Although little is said about the role of SDS in this process (except to lower the surface tension), I think this may in fact be the reason that this process works. SDS is commonly used as an additive in electrodeposition. It is known to inhibit the electrodeposition of materials - particularly on certain facets of the substrate. Is it possible that that SDS either blocks or at least inhibits growth on the regions between the particles and forces the growth to occur by transport of ions through the MOF to the electrode surface? In the absence of SDS, does this process work at all? This would still be an interesting mechanism, but it would be different than the authors suggest.

Reviewer #2 (Remarks to the Author):

The authors have made great efforts to address my previous concerns and the quality of this work has improved considerably. Particularly, additional experiments were performed to show the adjustability of this strategy as well as its broad application range. Furthermore, the plasmonic properties of the prepared Ag nanotriangle arrays and their applications in SERS sensing were explored. Impressively, the effects of the SDS were carefully elucidated by taking into account the chemical potentials as suggested. I would like to recommend acceptance of the manuscript for publication.

Reviewer #3 (Remarks to the Author):

In the revised manuscript titled "Metal Organic Framework Template-Guided Electrochemical Lithography on Substrates for SERS Sensing Applications" by Yang et al., notable improvements have been made that enhance the clarity and novelty of the study. The authors have thoughtfully addressed all of my previous comments, and I am pleased with the inclusion of supplementary experiments and in-depth discussions in this version.

Based on the revisions and additions, I have no further comments to make, and I confidently recommend the publication of this manuscript in its current form.

Reviewer #4 (Remarks to the Author):

I am happy with the changes made to the manuscript. The authors have carefully considered all criticism provided by the referees. I recommend the revised manuscript for publication in Nature Communications. No further changes are required.

Responses to Referee's comments

Reviewer #1:

Overall Comment: *The authors have greatly improved the paper by carefully addressing comments by all of the reviewers. I am still confused as to why metal ions would prefer to diffuse through the MOF instead of direct reduction at the metal surface between the MOF particles. I realize that transport of metal ions through the MOF can be relatively fast, but do they really transport faster in the pores than in the solution? Although little is said about the role of SDS in this process (except to lower the surface tension), I think this may in fact be the reason that this process works. SDS is commonly used as an additive in electrodeposition. It is known to inhibit the electrodeposition of materials - particularly on certain facets of the substrate. Is it possible that SDS either blocks or at least inhibits growth on the regions between the particles and forces the growth to occur by transport of ions through the MOF to the electrode surface? In the absence of SDS, does this process work at all? This would still be an interesting mechanism, but it would be different than the authors suggest.*

Author response: Thanks for your constructive comments. First of all, the middle part of the UiO-66 octahedra within the UiO-66 octahedron monolayer formed a compact solid surface with no pores (Fig. R1). To make it more clear, the schematic of the monolayer UiO-66 octahedron template and the monolayer spherical polystyrene (PS) spheres was illustrated in Fig. R2. Different from the conventional colloidal crystal template formed by PS spheres with an interstitial within three adjacent PS spheres, no through pores existed within the monolayer UiO-66 octahedron template. The solid surface separated the electrolyte solution above the top part of the octahedra from the bottom part (except for the defects within the monolayer octahedron template). As a result, the blocking plane at the middle part of the monolayer octahedron template prohibited the transport of silver ions via the solution route. The Ag⁺ ions have to enter the UiO-66 octahedra to reach the cathode electrode surface.

Fig. R1 Schematics of different sectional views of the monolayer UiO-66 octahedron film. From top to bottom: Cut from the middle of the UiO-66 octahedra, from a random position of the gaps and from the middle of the gaps, respectively.

Fig. R2 Schematics of the monolayer UiO-66 octahedron template without through pores (a) and the interstitials formed within the monolayer PS spheres (b).

Near-frictionless transport of ions have been observed in MOF nanochannels (for example, Nature 2023, 617, 299). In our experiments, we observed the shift of the adsorption equilibrium of the Ag^+ ions in the UiO-66@electrolyte system by the inductively coupled plasma-mass spectrometry (ICP-MS) measurements Fig. R3. 100 mg of UiO-66 octahedron powders were immersed in 1 ml of electrolyte solutions containing Ag^+ ions with a concentration of 0.3 M for 18 h. The concentration variation of Ag^+ ions over time was monitored by ICP-MS, which was compared to that of the electrolyte without UiO-66 octahedron powders. We observed a rapid decrease in the remaining concentration of Ag^+ ions in the solution initially and reached adsorption equilibrium after ~ 7 hours (Fig. R3a). This was referred to as the 1st adsorption equilibrium. Subsequently, we introduced additional Ag^+ ions to the solution to disrupt the equilibrium, and the process was repeated to reach the 2nd and 3rd equilibrium states. The remaining concentrations of Ag^+ ions after adsorption equilibrium were shown in Fig. R3b. The decrease in the concentration of Ag^+ ions in the solution before and after adsorption confirmed the entrance of Ag^+ ions into the nanochannels of UiO-66. The gradual decrease in the concentration difference of Ag^+ ions before and after adsorption can be attributed to the gradual occupancy of the nanochannels by Ag^+ ions.

Fig. R3 Ag^+ ion adsorption by UiO-66 octahedron powders. a, The concentration variation of the Ag^+ ions remaining in the solution after adsorption by the UiO-66 powders for different times; **b,** The concentration of Ag^+ ions remaining in the solution after introducing more Ag^+

ions into the equilibrium adsorption system. 1st, 2nd, and 3rd represent the remaining concentration of Ag⁺ ions in the solution before and after UiO-66 adsorption.

We further performed theoretical calculations of the chemical potential to understand the driving force of the Ag⁺ ions to enter the MOF nanochannels. The difference in chemical potentials ($\Delta\mu$) determines the mass transfer direction. Therefore, we calculated the $\Delta\mu$ of Ag⁺ ions between within UiO-66 nanochannels and in the bulk electrolyte solution (Fig. R4). The chemical potential (μ) of Ag⁺ ions is decomposed into the ideal term and the excess term (μ^{ex}):

$$\mu = RT\ln\Lambda^3\rho + \mu^{ex} \quad (1)$$

Where k_B is the Boltzmann constant; Λ is the de Broglie thermal wavelength of Ag; ρ is the number density of Ag⁺ ions. The difference in chemical potential ($\Delta\mu$) was then illustrated as:

$$\Delta\mu = k_B T \ln \frac{\rho_{UiO-66}}{\rho_{sol}} + \mu_{UiO-66}^{ex} - \mu_{Sol}^{ex} \quad (2)$$

We computed the μ_{UiO-66}^{ex} of -454.6 ± 2.3 kJ mol⁻¹ and μ_{Sol}^{ex} of -429.7 ± 0.1 kJ mol⁻¹ through molecular dynamics (MD) simulations combined with the Bennett acceptance ratio method (see Method section for details). The difference of $\mu_{Sol}^{ex} - \mu_{UiO-66}^{ex} = 24.9$ kJ mol⁻¹ indicates that Ag⁺ ions prefer to enter the nanochannels of UiO-66 until the ratio of $\frac{\rho_{UiO-66}}{\rho_{sol}}$ approaches approximately 1.2×10^4 , resulting in $\Delta\mu = 0$. However, it is important to consider the accommodation limit of Ag⁺ ions within UiO-66. Our MD simulations revealed that the population of Ag⁺ ions was highly concentrated at the eight tetrahedral cages and two octahedral cages in the unit cell of UiO-66 with a volume of 9.3 nm³, corresponding to the number density of ~ 1.1 nm⁻³. This information implies that when UiO-66 is immersed in the electrolyte, all the cages in the UiO-66 were occupied by Ag⁺ ions at the saturated density ρ_{UiO-66}^{sat} of ~ 1.1 nm⁻³. We calculated the number density of Ag⁺ ions from the adsorption equilibrium in Fig. R3. The saturated density, ρ_{UiO-66}^{sat} , was ~ 1.3 nm⁻³, in agreement with the predicted value by the MD simulations. Therefore, both the experimental and the simulation results suggest that all the cages within the UiO-66 were occupied by Ag⁺ ions when UiO-66 octahedra were immersed in the electrolyte used in our experiments.

Fig. R4 Schematic of the chemical potential difference of Ag⁺ ions between in the bulk electrolyte solution and in the nanochannels of the UiO-66.

As you said, SDS is commonly used as an additive to inhibit the growth of certain crystal planes during wet chemical or electrochemical fabrication processes. The Au substrate used in our experiments was polycrystalline fabricated by thermal evaporation. Therefore, the adsorption of SDS on the polycrystalline Au substrate should be random and unlikely to induce the formation of the Ag nanotriangle arrays after electrodeposition. Moreover, uniformly structured Ag nanofilms could be electrodeposited onto the Au substrate without the UiO-66 octahedron coverage in the electrolyte composed of SDS, indicating that SDS could not prohibit the growth of Ag during electrodeposition.

The main role of SDS is to promote the entrance of Ag⁺ ions into the nanochannels of UiO-66 octahedra. We will continue to explore the roles of SDS in our future work. Without SDS, neither Ag nanoframe arrays nor the Ag nanotriangle arrays could be electrodeposited as illustrated in Supplementary Figure 10.